# State Space Models on Temporal Graphs:
# A First-Principles Study

**Jintang Li**[1*]**, Ruofan Wu**[2*]**, Xinzhou Jin**[1]**, Boqun Ma**[3]**, Liang Chen**[1†]**, Zibin Zheng**[1]

[1]Sun Yat-sen University, [2]Coupang, [3]Shanghai Jiao Tong University

✉ {lijt55,jinxzh5}@mail2.sysu.edu.cn,{wuruofan1989,boqun.mbq}@gmail
{chenliang6,zhzibin}@mail.sysu.edu.cn}

## Abstract

Over the past few years, research on deep graph learning has shifted from static graphs to temporal graphs in response to real-world complex systems that exhibit dynamic behaviors. In practice, temporal graphs are formalized as an ordered sequence of static graph snapshots observed at discrete time points. Sequence models such as RNNs or Transformers have long been the predominant backbone networks for modeling such temporal graphs. Yet, despite the promising results, RNNs struggle with long-range dependencies, while transformers are burdened by quadratic computational complexity. Recently, state space models (SSMs), which are framed as discretized representations of an underlying continuous-time linear dynamical system, have garnered substantial attention and achieved breakthrough advancements in *independent* sequence modeling. In this work, we undertake a principled investigation that extends SSM theory to temporal graphs by integrating structural information into the online approximation objective via the adoption of a Laplacian regularization term. The emergent continuous-time system introduces novel algorithmic challenges, thereby necessitating our development of GRAPHSSM, a graph state space model for modeling the dynamics of temporal graphs. Extensive experimental results demonstrate the effectiveness of our GRAPHSSM framework across various temporal graph benchmarks.

## 1 Introduction

As a class of neural networks designed to operate directly on graph-structured data, graph neural networks (GNNs) [21, 43, 24] have achieved remarkable success and have established new state-of-the-art performance across a broad spectrum of graph-based learning tasks [19]. While significant progress has been made in researching *static* graphs, many real-world networks, such as social, traffic, and financial networks may exhibit *temporal* behaviors that carry valuable time information [20, 23]. This gives rise to temporal (dynamic) graphs, wherein the nodes and edges of the graph may undergo constant or periodic changes over time. In applications where temporal graphs arise, modeling and exploiting the dynamic nature of the continuously evolving graph is crucial in representing the underlying data and achieving high predictive performance [22, 44, 41].

Learning over temporal graphs is typically approached as a sequence modeling problem in which graph snapshots form a sequence [34]. This often involves challenges related to long graph sequences and scalability issues [25]. Recurrent neural networks (RNNs) [46, 4, 18] have historically dominated sequence modeling over the last years. However, they have long been plagued by poor capability in modeling long sequences due to rapid forgetting. This hampers their performance in temporal

---

[*]Equal contribution.
[†]Corresponding author.

38th Conference on Neural Information Processing Systems (NeurIPS 2024).

graphs that require a broader context or longer time window to capture relevant dependencies and patterns. Recently, the advancement of Transformers [42] has led to a shift in this paradigm, given their superior performance. Yet, Transformers also struggle with long sequence learning because the computational and memory complexity of self-attention is quadratically dependent on the sequence length. The overwhelming computation and memory requirements/costs associated with Transformers makes them less applicable in practical applications handling long-term sequences [51].

Recently, state space models (SSMs) have emerged as a powerful tool for sequence modeling [11, 13, 29, 40, 6, 10]. The salient characteristic that distinguishes state space models as particularly compelling is their conceptualization of sequential inputs as discrete observations from an underlying process evolving in continuous time, which naturally arises in scenarios such as speech processing [40] and time series analysis [49]. SSMs sustain a latent state throughout an input sequence and formulate state update equations through the discretization of an underlying linear dynamical system (LDS). Owing to their invariant state size, SSMs exhibit an efficient inferential time complexity, akin to that of RNNs. Simultaneously, they overcome the long-range modeling deficiencies inherent to RNNs through meticulous initializations of state matrices which are theoretically shown to achieve an optimal compression of history [11].

Temporal graphs often manifest as discrete snapshots capturing the evolution of an underlying graph that is inherently dynamic and continuous in nature [20]. In this context, the SSM methodology could be appropriated as a foundational primitive for temporal graph modeling. However, SSMs are predominantly architected towards independent sequence modeling. Hence, the task of systematically incorporating time-varying structural information into the SSM framework poses significant challenges. Specifically, it remains unexplored as to whether the foundational methodology of discretized LDS is readily applicable to the domain of temporal graphs.

In this work, we advance the SSM methodology to encompass temporal graphs from the first principles. Rather than presupposing the evolution of the underlying temporal graph, we dive into the fundamental problem of online function approximation that underpins the theoretical development of SSMs for sequence modeling [11]. By solving a novel Laplacian regularized online approximation objective, we derive a piecewise dynamical system that compresses historical information of temporal graphs. The piecewise nature of the obtained continuous-time system poses new challenges toward discretization into linear recurrences, thereby motivating our design of GRAPHSSM, a state space framework for temporal graphs. The main contributions of this work are summarized as follows:

- We introduce the GHIPPO abstraction, a novel construct predicated on the objective of Laplacian regularized online function approximation. This abstraction can alternatively be conceptualized as a memory compression primitive that simultaneously compresses both the feature dynamics and the evolving topological structure of the underlying temporal graph. The solution to GHIPPO is characterized by a dynamical system that is piecewise linear in node feature inputs.

- We introduce GRAPHSSM, a flexible state space framework designed for temporal graphs, which effectively addresses the key algorithmic challenge of unobserved graph mutations that impedes the straightforward discretization of the GHIPPO solution into (linear) recurrences through employing a novel mixed discretization strategy.

- Experimental results on six temporal graphs have validated the effectiveness of GRAPHSSM. In particular, GRAPHSSM has the advantages in scaling efficiency compared to existing state-of-the-arts, which can generalize to temporal graphs with long-range snapshots.

## 2 Related work

### 2.1 Temporal graph learning

A major branch of temporal graph learning methods consists of snapshot-based methods, which handle discrete-time temporal graphs by learning the temporal dependencies across a sequence of time-stamped graphs. Early works mainly focus on learning node representations by simulating temporal random walks [39] or modeling the triadic closure process [50] on multiple graph snapshots. These methods typically generate piecewise constant representations and may suffer from the staleness problem [20]. In recent years, the most established solution has been switched to combine sequence models (e.g., RNNs [46] and SNNs [38, 8]) with static GNNs to capture temporal dependencies and correlations between snapshots [47, 34, 39, 25]. To better translate the success achieved on static

graphs in both their design and training strategies, recent frameworks such as ROLAND [48] and its variants [53, 17] have been proposed to repurpose static GNNs to temporal graphs. There is another important line of research that focuses on continuous-time temporal graphs, we kindly refer readers to [27] and [20] for comprehensive surveys on this research topic.

## 2.2 State space models

State space models (SSMs) have historically served as a pivotal tool in fields such as signal processing [30] and time series analysis [3]. In recent advancements, they have also seen active adoption as a layer within neural sequence modeling frameworks [11, 13, 29, 40, 10]. The linear nature of SSMs confers several significant advantages. Key among these is the better-controlled stability that enables effective long-range modeling through careful initializations of state space layer parameters [11, 31], with the most representative method being HIPPO [11], a theory-driven framework notable for its optimal memory compression on continuous sequence inputs. Moreover, the computational efficacy of SSMs is notably enhanced through the use of techniques such as convolutions [13, 6] or parallel scans [40]. The promising properties of SSMs also attracts further explorations on graphs [2].

**Comparison.** The usual paradigms for designing sequence models over graphs involve recurrence (e.g. RNNs [46]), integrate-and-fire (e.g. SNNs [38, 8]), or attention (e.g. Transformers [42]), which each come with tradeoffs [14]. For example, RNNs are a natural recurrence model for sequential modeling that require only constant computation/storage per time step, but are slow to train and suffer from the rapid forgetting issue. This empirically limits their ability to handle long sequences. SNNs share a similar recurrent architecture with RNNs while using 1-bit spikes to transmit temporal information, which would sacrifice expressivity and potentially suffer from optimization difficulties (e.g., the "vanishing gradient problem") [26]. Transformers encode local context via attention mechanism and enjoy fast, parallelizable training, but are not sequential, resulting in more expensive inference and an inherent limitation on the context length. Compared to the aforementioned architectures, SSMs particularly the promising Mamba (S6) model [10], offer advantages such as fast training and inference, along with fewer parameters and comparable performance. These characteristics make SSMs particularly well-suited for sequence modeling, even (or especially) on extremely long sequences. Comparisons among these architectures are illustrated in table 1

Table 1: Comparisons of different neural network architectures for sequence modeling.

|  | RNNs [46, 4, 18] | SNNs [38, 8] | Transformers [42] | SSMs (S6 [10]) |
| --- | --- | --- | --- | --- |
| **Training** | Slow | Slow | Fast | Fast |
| **Inference** | Fast | Fast | Slow | Fast |
| **Para. Size** | Medium | Extremely small | Large | Small |
| **Performance** | ☆☆☆ | ☆☆☆ | ☆☆☆☆☆ | ☆☆☆☆ |
| **Limitations** | Forgetting | Vanishing gradients | Mem. & Time: $O(n^2)$ | ? |

# 3 The GRAPHSSM framework

The primary motivation of our framework is the fact that discrete-time temporal graphs are sequential observations of an underlying temporal graph that evolves continuously. Adopting this functional viewpoint, we will first develop a piecewise recurrent memory update scheme in section 3.1 that optimally approximates the underlying continuous-time temporal graph, utilizing a novel extension of the HIPPO abstraction to graph-typed inputs [11]. The proposed framework retains many nice properties of HIPPO while posing the new challenge of *unobserved graph mutation* when handling discretely-observed observations, which we analyze in section 3.2 and propose a mixing mechanism to improve the recurrent approximation. Finally, we present GRAPHSSM framework in section 3.3. An overview of GRAPHSSM is shown in figure 1.

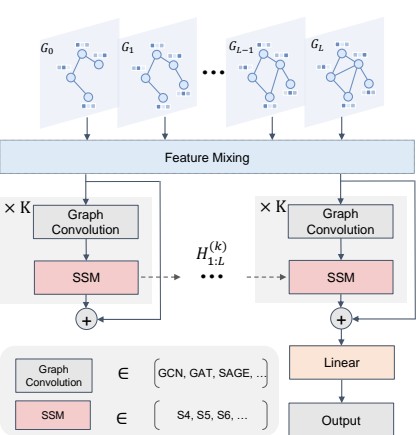

Figure 1: GRAPHSSM framework.

### 3.1 GHIPPO: HIPPO on temporal graphs

**Setup.** We fix a time interval $[0, T]$. A temporal graph on $[0, T]$ is characterized by two *processes* $G$ and $X$: For each $t \in [0, T]$, the process $G$ maps $t$ to a graph object $G(t) = (V(t), E(t))$. We assume the node process $V(t)$ to be fixed over time, i.e., $V(t) \equiv V, t \in [0, T]$ with $N_V = |V|$ and discuss the case for varying node processes in appendix B.2. The edge process $E(t)$ is a piecewise-constant process with a finite number $M$ of mutations over $[0, T]$ that are described via a sequence of *events*:

$$\mathscr{E} = \{\mathcal{E}_1, \dots, \mathcal{E}_M\} \text{ with each } \mathcal{E}_m = (u_m, v_m, t_m, a_m), 1 \leq m \leq M. \quad (1)$$

Each event $\mathcal{E}_m$ constitutes an interaction between node pair $(u_m, v_m)$ at time $t_m$ with action $a_m$, the action could be either insertion or deletion. The evolution process is thus depicted as the following:

$$G(0) \xrightarrow{\mathcal{E}_1} G(t_1) \xrightarrow{\mathcal{E}_2} G(t_2) \longrightarrow \cdots \longrightarrow G(t_{M-1}) \xrightarrow{\mathcal{E}_M} G(t_M) = G(T). \quad (2)$$

The process $X$ maps $t$ to a node feature matrix $X(t) \in \mathbb{R}^{N_V \times d}$ with feature dimension $d$. Throughout this paper, it is often helpful to view $G$ and $X$ as graph-valued and matrix-valued *functions*. In typical discrete-time temporal graph learning problems, the underlying graph is observed at timestamps $\tau_1, \dots, \tau_L$ with time gaps $\Delta_l = \tau_l - \tau_{l-1}, 2 \leq l \leq L$. The observations thus form a sequence of snapshots $\{G(\tau_l), X(\tau_l)\}_{1 \leq l \leq L}$ which are abbreviated as $\{G_{1:L}, X_{1:L}\}$. Notably, the observation times are usually *interleaved with* the mutation times, resulting in the majority of mutation times remain unobserved. This situation presents significant challenges in effectively modeling the dynamics of graph evolution, a topic that will be further explored subsequently.

**The HIPPO abstraction.** Algorithmically, the goal of continuous-time dynamic modeling is to design a *memory module* that optimally compresses all the historical information [11]. Under the context of univariate sequence modeling, the HIPPO framework [11] formalizes the memory compression problem into an online approximation problem in some function space and derives HIPPO operators under specific types of basis functions, among which the HIPPO-LEGS configuration has become the state-of-the-art in state-space sequence modeling paradigms [13, 40]. However, naively extending HIPPO abstraction to graph learning scenarios (via treating node features as inputs) could be deemed inadequate since HIPPO handles distinct inputs *independently*, without the capability to incorporate the interconnectivity information among various inputs which could potentially enhance the efficiency of memory compression. For illustrative purposes, in instances where input observations are noisy, the exploitation of neighborhood information has the potential to facilitate a denoising step, as evidenced in image processing applications [33] and semi-supervised learning primitives [52, 45]. To systematically utilize the connectivity information, we propose a new approximation paradigm, the *Laplacian-regularized online approximation* that extends HIPPO to graph modeling frameworks. Formally, we start with the simple setup with $d = 1$, i.e., each node possesses a scalar feature, and we propose an approximation scheme that simultaneously approximates the history of all the $N_V$ inputs up until time $t$, i.e., $\{X(s), s \in [0, t]\}$ using their corresponding memories at time $t$, i.e., $Z(t) = \{z_v(t)\}_{v \in V} \in \mathbb{R}^{N_V \times 1}$ according to the following objective at time $t$:

$$\mathcal{L}_t(Z; G, X, \mu) = \int_0^t \|X(s) - Z(s)\|_2^2 \, d\mu_t(s) + \alpha \int_0^t Z(s)^\top L(s) Z(s) d\mu_t s. \quad (3)$$

Here $\alpha > 0$ is a balancing constant, $\mu_t$ is a time-dependent measure that is supported on the interval $[0, t]$ which controls the importance of various parts of the input domain[3] and $L(t)$ is a normalized Laplacian at time $t$, which allows definition such as the symmetric normalized Laplacian $L_{\text{sym}}(s) = I - D(s)^{-1/2} A(s) D(s)^{-1/2}$ where $D(s)$ is a diagonal matrix whose diagonals are node degrees, or random walk normalized Laplacian $L_{\text{rw}}(s) = I - D(s)^{-1} A(s)$. The objective (3) is understood as the ordinary HIPPO approximation objective augmented with a regularization component that encourages the *smoothness* of memory compression with respect to adjacent nodes. [4] The imposition of smoothness constraints commonly emerges as a beneficial relational inductive bias in the context of graph learning [1]. By leveraging the data from adjacent nodes, one can potentially achieve a more effective denoising effect during the process of node memory compression. To specify

---

[3]Technically, we require $Z$ and $X$ to reside within some appropriately defined Hilbert space. A comprehensive treatment will be provided in appendix C.

[4]To be more precise, at any time $s \in [0, t]$, the integrand $Z(s)^\top L(s) Z(s)$ inside the second term of (3) attains its minimum when $Z(s)$ satisfies certain smoothness criterion which is determined via the choice of graph Laplacian. A more detailed explanation is deferred to appendix B.1.

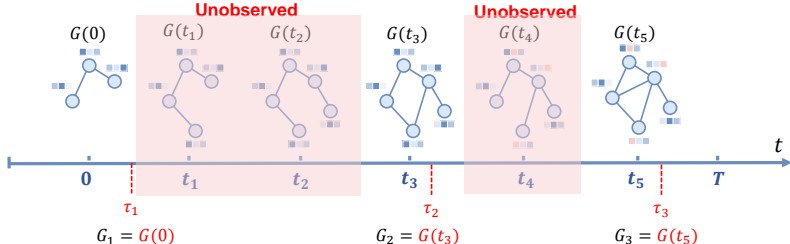

Figure 2: Illustrative example of the *unobserved graph mutation* issue. In this example, the underlying graph is observed at time points $\tau_1, \tau_2, \tau_3$ with two unobserved mutations between $[\tau_1, \tau_2)$ and one between $[\tau_2, \tau_3)$. These unobserved mutations result in temporal dynamics that are inconsistent across the observed intervals, thereby complicating direct applications of ODE discretization methods such as the Euler method or the zero-order hold (ZOH) method.

a suitable approximation subspace for memories $Z$, we adopt the approach of HIPPO that uses some $N$-dimensional subspace of polynomials which we denote as $\mathcal{P}_N$. Now we define a *graph memory projection operator* GPROJ$_t$ that maps the temporal graph up until time $t$ to a collection of $N_V$ polynomials with each one lies in $\mathcal{P}_N$, i.e.,

$$\text{GPROJ}_t (G, X) = \arg\min_{Z:z_v \in \mathcal{P}_N \ \forall v \in V} \mathcal{L}_t(Z; G, X, \mu). \qquad (4)$$

We further define a *coefficient* operator COEF$_t$ that maps each polynomial in the collection in (4) to the coefficients of the basis of orthogonal polynomials defined with respect to $\mu_t$, the following definition formalizes our extension of HIPPO to continuous-time temporal graphs which we term GHIPPO:

**Definition 1** (GHIPPO)**.** *Given a continuous-time temporal graph $(G, X)$, a time-varying measure family $\mu_t$, an $N$-dimensional subspace of polynomials $\mathcal{P}_N$, the GHIPPO operator at time $t$ is the composition of GPROJ$_t$ and COEF$_t$ that maps the temporal graph and node features to a collection of projection coefficients $U(t) \in \mathbb{R}^{N_V \times N}$, or GHIPPO$(G, X) = \text{COEF}_t(\text{GPROJ}_t(G, X))$.*

The most favorable property of the HIPPO framework on independent inputs is that the outputs of HIPPO operators are characterized via a concise ordinary differential equation (ODE) that takes the form of a linear time-invariant state space model (LTI-SSM). The following theorem states that most of the desirable properties of HIPPO are retained by GHIPPO except for the LTI property:

**Theorem 1.** *Let $G$ evolve according to (2). Taking $\mu_t$ to be the scaled Legendre measure (LegS) with $\mu_t = \frac{1}{t}\mathbb{I}_{[0,t]}$ where $\mathbb{I}_{[0,t]}$ stands for the indicator function of the interval $[0, t]$, the evolution of the outputs of GHIPPO operator is characterized by $M$ ODEs according to mutation times as follows:*

$$\frac{dU(t)}{dt} = U(t)A^\top + (I + \alpha L(t))^{-1} X(t) B^\top, \quad 1 \leq m \leq M, t \in [t_{m-1}, t_m) \qquad (5)$$

*where $A \in \mathbb{R}^{N \times N}$ and $B \in \mathbb{R}^{N \times 1}$ takes the same form as in the HIPPO formulation [11]:*

$$A_{nk} = -\begin{cases} \sqrt{(2n+1)(2k+1)} & \text{if } n > k, \\ n+1 & \text{if } n = k, \\ 0 & \text{if } n < k, \end{cases} \quad \text{and} \quad B_n = \sqrt{2n+1}, 1 \leq n \leq N. \qquad (6)$$

According to theorem 1, the solution (5) is LTI over each interval $[t_m, t_{m+1})$ during which the graph structure remains fixed. This property further extends to a piecewise LTI perspective over the interval $[0, T]$. Moreover, we may view the solution (5) as a two-stage procedure that could be intuitively described as *diffuse-then-update*. Specifically, this procedure entails a sequential execution, wherein an initial diffusion operation is applied to the features of the input nodes, succeeded by an update to the memory of these nodes.

## 3.2 Unobserved graph mutations and mixed discretization

Theorem 1 establishes an analogue of HIPPO theory on temporal graphs. It is straightforward to verify that most of the subsequent refinements of HIPPO apply to GHIPPO as well. Among

these we will utilize the popular technique of *diagonal state spaces* [15, 12] that simply sets $A$ as a diagonal matrix with negative diagonal elements[5]. To apply the GHiPPO framework to discrete-time temporal graphs, a critical step is to develop a discretized version of (5). However, unlike ordinary HiPPO where we can use standard discretization techniques of ODEs to discretize LTI equations, the GHiPPO ODE contains discontinuities that correspond to mutation times of the underlying temporal graph, which are often not observed given only access to a list of snapshots. This issue of *unobserved dynamics* complicates the development of a viable discretization scheme for GHiPPO, as is pictorially illustrated in figure 2. To devise a solution to this challenge, we start by analyzing a hypothetical *oracle scenario* in which all mutations are observable.

**An oracle discretization.** We consider a time range $[\tau_{l-1}, \tau_l)$ between the $l-1$th and the $l$th snapshot, and assume there are altogether $M_l$ mutation events $\{\mathcal{E}_{l,i}\}_{1 \leq i \leq M_l}$ happened during this period. Let $G_{l,0} = G_{l-1}$ be the graph snapshot at $\tau_{l-1}$, the following process describes the structural evolution inside the interval $[\tau_{l-1}, \tau_l)$:

$$G_{l-1} = G_{l,0} \xrightarrow{\mathcal{E}_{l,1}} G_{l,1} \xrightarrow{\mathcal{E}_{l,2}} G_{l,2} \longrightarrow \cdots \longrightarrow G_{l,M_l-1} \xrightarrow{\mathcal{E}_{l,M_l}} G_{l,M_l} = G_l \tag{7}$$

Next, we derive a discretization formula under the strategy of zeroth-order-hold (ZOH). We assume that all intermediate mutations are observed, with the node features staying fixed between mutations, i.e., $X(t) \equiv X_{l,i}, t \in [t_{l,i-1}, t_{l,i})$. The following theorem characterizes the resulting state evolution:

**Theorem 2** (Oracle discretization of (5))**.** *Assume $A$ is a diagonal matrix with negative diagonals, for any $1 \leq l \leq L$. Let $L_{l,i}$ be some Laplacian of $G_{l,i}$, we have the following oracle update rule:*

$$U_l = U_{l-1}e^{\Delta_l A} + \widetilde{X}_l \left(e^{\Delta_l A} - I\right) A^{-1}, \ \widetilde{X}_l = \sum_{i=0}^{M_l}(I + \alpha L_{l,i})^{-1}X_{l,i}\Lambda_i B^\top, \tag{8}$$

*where $U_l \in \mathbb{R}^{N_V \times N}$ denotes the discretized state at step $l$ with $U_0 = 0$. For each $1 \leq l \leq L, 0 \leq i \leq M_l$, $\Lambda_i \in \mathbb{R}^{N \times N}$ are non-negative diagonal matrices with values depending only on the mutation times, which satisfy $\sum_{i=0}^{M_l} \Lambda_i = I$.*

**Mixed discretization.** According to (8), given all the (unobserved) mutation information, the state update rule is equivalent to applying ZOH to $\widetilde{X}_l$ which is an *element-wise convex combination* of all the diffused node features. In practice, among all the components of $\widetilde{X}$, we only have access to $X_{l-1}, X_l, G_{l-1}, G_l$ with the rest left unobserved. Therefore, we propose *mixed discretization* as an approach to approximate $\widetilde{X}_l$. Specifically, we introduce the following mechanisms:

$$\widehat{X}_l^{(\text{O})} = \text{GNN}_\theta\left(X_l, G_l\right), \hspace{5em} \text{(ordinary ZOH)}$$

$$\widehat{X}_l^{(\text{F})} = \text{GNN}_\theta\left(\text{Mix}_\phi\left(X_{l-1}, X_l\right), G_l\right), \hspace{3.5em} \text{(feature mixing)}$$

$$\widehat{X}_l^{(\text{R})} = \text{Mix}_\phi\left(\text{GNN}_\theta\left(X_{l-1}, G_{l-1}\right), \text{GNN}_\theta\left(X_l, G_l\right)\right), \hspace{1em} \text{(representation mixing)}$$

which are compositions of inter-node mixing (a consequence of diffusion) and intra-node mixing (mixing node features of consecutive snapshots). For the process of inter-node mixing, we opt to approximate the diffusion operation with a learnable shallow graph neural network (typically a 1-layer GNN) parameterized by $\theta$ to alleviate the computation burden and improve flexibility[6]. A detailed discussion considering the relation between certain GNN formulations and the choice of Laplacian is presented in appendix B.1. In the context of intra-node mixing, we introduce a Mix module parameterized by $\phi$ to merge either consecutive node features (as illustrated in (feature mixing)) or consecutive node representations produced by the GNN model (as illustrated in (representation mixing)). In this paper, we assess two simple Mix instantiations: Convolution with a kernel size of 2 (Conv1D) and a gating mechanism that interpolates between the two inputs (Interp). We postpone a comprehensive description of the mixing methods to appendix D.1. The resulting discretized system is presented as the following matrix-valued state space model:

$$\begin{aligned} U_l &= U_{l-1}e^{\Delta_l A} + \Delta_l \widehat{X}_l^{(\cdot)} B^\top \\ Y_l &= U_l C^\top. \end{aligned} \quad \text{with} \quad \widehat{X}_l^{(\cdot)} \in \left\{\widehat{X}_l^{(\text{O})}, \widehat{X}_l^{(\text{F})}, \widehat{X}_l^{(\text{R})}\right\}, 1 \leq l \leq L. \tag{9}$$

---

[5]More concretely, diagonal SSMs are defined by diagonal $A$ matrices whose diagonal elements lie on the complex plane with negative real parts [12], yet recent developments have found that complex state matrices are often not necessary [10]. For ease of representation, we only explore real state matrices in this paper.

[6]We let the balancing constant $\alpha$ be absorbed into the learnable parameters. Indeed, for GNNs that employ asymmetric aggregation [16], it is plausible to conceptualize the GNN as engaging a form of auto-balancing.

When exact timestamps for snapshots are unavailable, we use the adaptive time step strategy as in [11, 10] that models $\Delta$ a 1-dimensional affine projection of the inputs followed by a non-negative activation like softplus. Finally, we utilize the approximation $A^{-1}\left(e^{\Delta A} - I\right) \approx \Delta I$ for diagonal $A$s, and equip the system with an output $Y$ with a state projection matrix $C \in \mathbb{R}^{N \times 1}$.

### 3.3 The GRAPHSSM framework

Having established the SSM equation (9), we are ready to introduce our main framework GRAPHSSM. In alignment with conventional design paradigms in the SSM literature, we define a depth-$K$ GRAPHSSM model through the sequential composition of $K$ GRAPHSSM blocks, with each block characterized as follows:

$$H_{1:L}^{(k)} = \sigma\left(\text{SSMLAYER}\left(H_{1:L}^{(k-1)}, G_{1:L}\right)\right) + \text{LINEAR}\left(H_{1:L}^{(k-1)}\right), 1 \leq k \leq K, \qquad (10)$$

where we use $H_{1:L}^{(k)}$ to denote the concatenation of the hidden representation at depth $k$ of all the snapshots along the sequence dimension and $H_{1:L}^{(0)}$ are the node features $X_{1:L}$. The GRAPHSSM blocks, as outlined in (10), incorporate an SSM layer that operates on graph snapshot inputs. This is followed by the application of a nonlinear activation $\sigma$ and the integration of a residual connection which we denote as the addition of a linear projection of inputs with Linear denotes a linear projection layer that ensures dimension compatibility.

**GRAPHSSM-S4.** The architectural formulation of the SSM layer essentially involves the expansion of the one-dimensional recurrence, as specified in (9), to accommodate general dimensions, i.e., $d > 1$. This expansion is achieved in a straightforward manner by utilizing an individual SSM for each dimension. Consequently, the emergent SSM layer adopts a Single-Input, Single-Output (SISO) configuration. Such a design is intuitively understood as the graph learning analogue of S4 [13], which we term GRAPHSSM-S4.

**GRAPHSSM-S5 and GRAPHSSM-S6.** In addition to the SISO implementation, we further introduce two variants within the GRAPHSSM framework. The first alternative represents a Multiple-Input, Multiple-Output (MIMO) extension of (9), wherein a single SSM system is applied across all dimensions. This variant serves as a graph-informed analogue to the S5 model [40]. The second variant extends the S4 model by facilitating input-controlled time intervals and state matrices ($\Delta$, $B$, and $C$). This innovation yields a selective state space model, drawing parallels to the latest SSM architectures such as S6 [10].

A detailed exposition of the GRAPHSSM-S4 (resp. GRAPHSSM-S5, GRAPHSSM-S6) layer is provided in algorithm 1 (resp. algorithm 2, algorithm 3) in appendix D.2. The overall end-to-end architecture is briefly illustrated in figure 1, where we use feature mixing as the mixing mechanism for illustration.

**Remark 1** (Choice of mixing mechanisms). *In the GRAPHSSM architecture, each SSM layer incorporates a mixing mechanism. Based on our empirical investigations, we have observed that employing more sophisticated mixing strategies such as (feature mixing) and (representation mixing), yields benefits predominantly when these are applied exclusively to the lowermost layer. Specifically, this entails utilizing either $\widehat{X}_l^{(F)}$ or $\widehat{X}_l^{(R)}$ configurations in the initial layer, while defaulting to $\widehat{X}_l^{(O)}$ for the layers that follow. An intuitive rationale behind this strategic layer-specific choice will be elucidated in appendix B.3.*

## 4 Experiments

This section presents our key experimental findings on the temporal node classification task. Also, ablation studies of the key design choices are presented. Due to space limitation, the detailed experimental settings are deferred to appendix F.

### 4.1 Experimental results

**Node classification performance.** The node classification performance of all methods is presented in table 2. It has been observed that graph embedding methods, especially static ones, tend to underperform in most cases. This is expected since these methods are typically trained in an

Table 2: Node classification performance (%) on four small scale temporal graphs. The best and the second best results are highlighted as **red** and **blue**, respectively.

| | DBLP-3 | | Brain | | Reddit | | DBLP-10 | |
|---|---|---|---|---|---|---|---|---|
| | **Micro-F1** | **Macro-F1** | **Micro-F1** | **Macro-F1** | **Micro-F1** | **Macro-F1** | **Micro-F1** | **Macro-F1** |
| DeepWalk [36] | $47.53_{\pm 0.4}$ | $47.21_{\pm 0.2}$ | $51.45_{\pm 0.6}$ | $51.03_{\pm 0.8}$ | $26.82_{\pm 0.6}$ | $26.75_{\pm 0.4}$ | 66.38 | 67.12 |
| Node2Vec [9] | $48.79_{\pm 0.3}$ | $48.42_{\pm 0.4}$ | $53.51_{\pm 0.5}$ | $52.95_{\pm 0.6}$ | $25.47_{\pm 0.6}$ | $25.44_{\pm 0.5}$ | 67.31 | 66.93 |
| HTNE [54] | $48.98_{\pm 0.2}$ | $48.74_{\pm 0.3}$ | $22.31_{\pm 0.8}$ | $22.12_{\pm 0.5}$ | $26.96_{\pm 0.5}$ | $26.80_{\pm 0.7}$ | 68.79 | 68.36 |
| M$^2$DNE [28] | $49.12_{\pm 0.5}$ | $48.87_{\pm 0.4}$ | $23.79_{\pm 0.4}$ | $23.54_{\pm 0.6}$ | $25.79_{\pm 0.6}$ | $25.61_{\pm 0.4}$ | 69.71 | 69.75 |
| DynamicTriad [50] | $48.78_{\pm 0.5}$ | $48.63_{\pm 0.6}$ | $21.71_{\pm 0.7}$ | $21.94_{\pm 0.7}$ | $28.76_{\pm 0.5}$ | $28.51_{\pm 0.5}$ | 66.95 | 66.42 |
| MPNN [32] | $81.78_{\pm 0.6}$ | $81.46_{\pm 1.2}$ | $90.97_{\pm 1.4}$ | $91.01_{\pm 1.5}$ | $40.85_{\pm 1.3}$ | $40.64_{\pm 1.2}$ | $67.74_{\pm 0.3}$ | $65.05_{\pm 0.5}$ |
| STAR [47] | $\textbf{\textcolor{blue}{84.74}}_{\pm 1.0}$ | $\textbf{\textcolor{blue}{84.20}}_{\pm 1.2}$ | $92.08_{\pm 1.3}$ | $92.23_{\pm 1.3}$ | $43.42_{\pm 2.3}$ | $43.43_{\pm 2.4}$ | $72.98_{\pm 1.5}$ | $72.03_{\pm 1.2}$ |
| tNodeEmbed [39] | $84.51_{\pm 1.2}$ | $83.57_{\pm 1.1}$ | $\textbf{\textcolor{blue}{92.35}}_{\pm 0.8}$ | $\textbf{\textcolor{blue}{92.30}}_{\pm 1.0}$ | $42.11_{\pm 1.8}$ | $42.06_{\pm 1.3}$ | $74.19_{\pm 1.8}$ | $74.23_{\pm 2.2}$ |
| EvolveGCN [34] | $84.01_{\pm 1.5}$ | $83.12_{\pm 1.5}$ | $92.20_{\pm 1.3}$ | $92.00_{\pm 1.0}$ | $41.24_{\pm 1.3}$ | $41.11_{\pm 1.5}$ | $71.32_{\pm 0.5}$ | $71.20_{\pm 0.7}$ |
| SpikeNet [25] | $83.92_{\pm 1.5}$ | $83.04_{\pm 1.1}$ | $92.00_{\pm 1.2}$ | $91.97_{\pm 1.2}$ | $40.42_{\pm 2.0}$ | $40.20_{\pm 2.1}$ | $74.86_{\pm 0.5}$ | $74.65_{\pm 0.5}$ |
| ROLAND [48] | $84.21_{\pm 1.4}$ | $84.06_{\pm 1.5}$ | $92.14_{\pm 1.2}$ | $91.85_{\pm 1.1}$ | $\textbf{\textcolor{blue}{44.22}}_{\pm 2.2}$ | $\textbf{\textcolor{blue}{44.25}}_{\pm 1.9}$ | $\textbf{\textcolor{blue}{75.01}}_{\pm 1.1}$ | $\textbf{\textcolor{blue}{74.98}}_{\pm 1.0}$ |
| GRAPHSSM | $\textbf{\textcolor{red}{85.26}}_{\pm 0.9}$ | $\textbf{\textcolor{red}{85.00}}_{\pm 1.3}$ | $\textbf{\textcolor{red}{93.52}}_{\pm 1.0}$ | $\textbf{\textcolor{red}{93.54}}_{\pm 0.9}$ | $\textbf{\textcolor{red}{49.21}}_{\pm 0.5}$ | $\textbf{\textcolor{red}{49.05}}_{\pm 0.7}$ | $\textbf{\textcolor{red}{76.80}}_{\pm 0.3}$ | $\textbf{\textcolor{red}{76.00}}_{\pm 0.4}$ |

unsupervised manner, solely focusing on exploiting the graph structure. We note that continuous-time methods HTNE and M$^2$DNE exhibit poor performance in DBLP-3, Brain, and Reddit even when compared to static methods. This indicates that continuous-time methods are not well-suited for handling discrete-time graphs, particularly in the absence of temporal continuity. As can also be observed from table 2, most temporal graph neural networks demonstrate good performance on DBLP-3 and Brain datasets, where the node labels are largely dominated by node attribute information [47]. However, for datasets like Reddit and DBLP-10, where graph topology information plays a more significant role in classification, the performance has notably degraded. This indicates that the baseline methods struggle to effectively capture the underlying evolving graph structure and exploit it for accurate classification. In contrast, our most performant architecture, GRAPHSSM-S4, exhibits an average improvement of 14% and 2% in Micro-F1 and Macro-F1 scores, respectively, compared to state-of-the-art baselines on the Reddit and DBLP-10 datasets. In addition, GRAPHSSM-S4 is a more preferable choice for long graph sequences, achieving new state-of-the-art performance on the DBLP-10 dataset.

**Scalability to large temporal graphs.** To explore the effectiveness of GRAPHSSM on large-scale and long-range temporal graphs, we conduct comparison experiments on arXiv and Tmall and present the result in table 3. Since both datasets exhibit a relatively high level of temporal continuity in the observed graph sequence, several advanced baselines have achieved good performance. However, the graph scale and long sequence still pose significant challenges for learning over both datasets, where most methods are insufficient to effectively and efficiently capture the long-range graph dynamics. In contrast, by leveraging the linear efficiency and long-range modeling capability of SSMs, GRAPHSSM outperforms strong baselines on both datasets.

Table 3: Node classification performance (%) on large scale temporal graphs. OOM: out-of-memory.

| | arXiv | | Tmall | |
|---|---|---|---|---|
| | **Micro-F1** | **Macro-F1** | **Micro-F1** | **Macro-F1** |
| DeepWalk [36] | $66.54_{\pm 0.3}$ | $43.01_{\pm 0.3}$ | 57.88 | 49.53 |
| Node2Vec [9] | $67.71_{\pm 0.5}$ | $43.69_{\pm 0.4}$ | 60.66 | 54.58 |
| HTNE [54] | $65.48_{\pm 0.3}$ | $42.25_{\pm 0.3}$ | 62.64 | 54.93 |
| M$^2$DNE [28] | $66.91_{\pm 0.5}$ | $43.52_{\pm 0.6}$ | 64.65 | 58.47 |
| DynamicTriad [50] | $61.10_{\pm 0.2}$ | $38.25_{\pm 0.3}$ | 60.72 | 51.16 |
| MPNN [32] | $64.68_{\pm 1.7}$ | $41.22_{\pm 1.5}$ | $58.07_{\pm 0.6}$ | $50.27_{\pm 0.5}$ |
| STAR [47] | OOM | OOM | OOM | OOM |
| tNodeEmbed [39] | OOM | OOM | OOM | OOM |
| EvolveGCN [34] | $65.17_{\pm 1.4}$ | $43.01_{\pm 1.3}$ | $61.77_{\pm 0.6}$ | $55.78_{\pm 0.6}$ |
| SpikeNet [25] | $66.69_{\pm 0.9}$ | $43.96_{\pm 1.0}$ | $\textbf{\textcolor{blue}{66.10}}_{\pm 0.3}$ | $\textbf{\textcolor{blue}{62.40}}_{\pm 0.6}$ |
| ROLAND [48] | $\textbf{\textcolor{blue}{68.27}}_{\pm 1.2}$ | $\textbf{\textcolor{blue}{48.01}}_{\pm 1.3}$ | OOM | OOM |
| GRAPHSSM | $\textbf{\textcolor{red}{70.67}}_{\pm 0.7}$ | $\textbf{\textcolor{red}{49.97}}_{\pm 0.5}$ | $\textbf{\textcolor{red}{66.29}}_{\pm 0.1}$ | $\textbf{\textcolor{red}{62.57}}_{\pm 0.1}$ |

Table 4: Node classification performance (%) with different SSM architectures.

| | DBLP-3 | | Brain | | Reddit | | DBLP-10 | |
|---|---|---|---|---|---|---|---|---|
| | **Micro-F1** | **Macro-F1** | **Micro-F1** | **Macro-F1** | **Micro-F1** | **Macro-F1** | **Micro-F1** | **Macro-F1** |
| GRAPHSSM-S4 | $\textbf{\textcolor{blue}{85.26}}_{\pm 0.9}$ | $\textbf{\textcolor{blue}{85.00}}_{\pm 1.3}$ | $93.52_{\pm 1.0}$ | $93.54_{\pm 0.9}$ | $\textbf{\textcolor{red}{49.21}}_{\pm 0.5}$ | $\textbf{\textcolor{red}{49.05}}_{\pm 0.7}$ | $\textbf{\textcolor{red}{76.80}}_{\pm 0.3}$ | $\textbf{\textcolor{red}{76.00}}_{\pm 0.4}$ |
| GRAPHSSM-S5 | $84.32_{\pm 1.5}$ | $83.57_{\pm 1.9}$ | $92.20_{\pm 0.6}$ | $92.05_{\pm 0.7}$ | $\textbf{\textcolor{blue}{44.75}}_{\pm 0.4}$ | $\textbf{\textcolor{blue}{44.79}}_{\pm 0.4}$ | $73.19_{\pm 0.6}$ | $72.95_{\pm 0.4}$ |
| GRAPHSSM-S6 | $\textbf{\textcolor{red}{85.74}}_{\pm 0.5}$ | $\textbf{\textcolor{red}{85.23}}_{\pm 0.6}$ | $\textbf{\textcolor{red}{93.80}}_{\pm 0.3}$ | $\textbf{\textcolor{red}{94.47}}_{\pm 0.6}$ | $42.52_{\pm 0.9}$ | $41.73_{\pm 1.1}$ | $\textbf{\textcolor{blue}{75.26}}_{\pm 0.3}$ | $\textbf{\textcolor{blue}{74.81}}_{\pm 0.2}$ |

**SSM architectures.** As GRAPHSSM is a general framework that generalizes SSMs to temporal graphs, we conduct experiments on extending GRAPHSSM with different ad-hoc SSMs, including S5 [40] and S6 [10]. The node classification results on four datasets are shown in table 4. By comparing different variants of GRAPHSSM, we can find that S4 is the best architecture for learning over temporal graph sequences. S5, being a simplified version of S4 with fewer parameters, achieves poor performance on all datasets. Notably, while S6 shows impressive performance in other modalities such as language or images [10, 51], it is observed that they underperform when applied to graph sequences. This indicates that the selective mechanism may not be a good fit for graph data.

Table 5: Ablation results (%) of GRAPHSSM-S4 with different mixing configurations.

| | DBLP-3 | | Brain | | Reddit | | DBLP-10 | |
| --- | --- | --- | --- | --- | --- | --- | --- | --- |
| | Micro-F1 | Macro-F1 | Micro-F1 | Macro-F1 | Micro-F1 | Macro-F1 | Micro-F1 | Macro-F1 |
| $\widehat{X}_1^{(O)} + \widehat{X}_2^{(O)}$ | $84.51_{\pm 0.9}$ | $84.28_{\pm 0.9}$ | $91.56_{\pm 1.1}$ | $91.99_{\pm 0.7}$ | $48.05_{\pm 2.8}$ | $47.99_{\pm 3.0}$ | $75.62_{\pm 0.5}$ | $74.65_{\pm 0.6}$ |
| $\widehat{X}_1^{(F)} + \widehat{X}_2^{(O)}$ | $\mathbf{85.12}_{\pm 0.5}$ | $\mathbf{84.82}_{\pm 0.3}$ | $\mathbf{92.36}_{\pm 0.8}$ | $92.54_{\pm 0.9}$ | $\mathbf{49.06}_{\pm 1.9}$ | $\mathbf{49.06}_{\pm 1.8}$ | $\mathbf{76.67}_{\pm 0.6}$ | $\mathbf{75.95}_{\pm 0.7}$ |
| $\widehat{X}_1^{(R)} + \widehat{X}_2^{(O)}$ | $84.98_{\pm 1.1}$ | $84.79_{\pm 1.0}$ | $\mathbf{93.52}_{\pm 1.0}$ | $\mathbf{93.54}_{\pm 0.9}$ | $\mathbf{49.21}_{\pm 0.5}$ | $\mathbf{49.05}_{\pm 0.7}$ | $\mathbf{77.76}_{\pm 0.5}$ | $\mathbf{77.54}_{\pm 0.6}$ |
| $\widehat{X}_1^{(O)} + \widehat{X}_2^{(R)}$ | $\mathbf{85.26}_{\pm 0.9}$ | $\mathbf{85.00}_{\pm 1.3}$ | $91.84_{\pm 1.9}$ | $91.88_{\pm 1.7}$ | $47.88_{\pm 1.8}$ | $47.94_{\pm 1.8}$ | $75.41_{\pm 0.7}$ | $74.89_{\pm 1.0}$ |

**Mixing mechanism.** We assess the effectiveness of various mixing mechanisms introduced in section 3.2 through a series of experiments conducted using the S4 variant of GRAPHSSM. The analysis spans four distinct configurations: no intra-node mixing ($\widehat{X}_1^{(O)} + \widehat{X}_2^{(O)}$), feature mixing at the first layer ($\widehat{X}_1^{(F)} + \widehat{X}_2^{(O)}$), and representation mixing at either the first ($\widehat{X}_1^{(R)} + \widehat{X}_2^{(O)}$) or second ($\widehat{X}_1^{(O)} + \widehat{X}_2^{(R)}$) layers. The findings, presented in table 5, indicate that the integration of the MIX module at the first layer generally leads to enhanced model performance. An intuitive explanation for this observed phenomenon is elaborated in appendix B.3.

**Initialization strategy.** Recent advancements have highlighted the crucial role of initialization in SSMs [12], prompting our investigation into the effects of various initialization strategies for the $A$ matrix. Specifically, we explore "hippo", "constant", and "random" initializations, with their comprehensive definitions provided in appendix D.2. The result, as shown in figure 3 exhibits distinct performance variations across different initialization strategies, with HIPPO being typically the dominant one which corroborates our theoretical motivations.

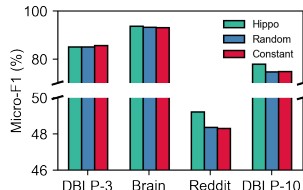

Figure 3: Comparison of GRAPHSSM with different initialization strategies.

## 5 Conclusion

In this work, we introduce a conceptualized GHIPPO abstraction on temporal graphs. Building upon GHIPPO, we propose GRAPHSSM, a theoretically motivated state space framework for modeling temporal graphs derived from a novel memory compression scheme. The proposed framework is computationally efficient and versatile in its design, which is further corroborated by strong empirical performance across various benchmark datasets. We also point out the unobserved graph mutation issue in temporal graphs and propose different mixing mechanisms to ensure temporal continuity across consecutive graph snapshots. Despite the promising results, the applicability of GRAPHSSM is presently confined to discrete-time temporal graphs. A discussion of our framework's current limitations and the scope for future extensions is presented in appendix E.

## Acknowledgement

The research is supported by the National Key R&D Program of China under grant No. 2022YFF0902500, the Guangdong Basic and Applied Basic Research Foundation, China (No. 2023A1515011050), Shenzhen Science and Technology Program (KJZD20231023094501003), and Tencent AI Lab RBFR2024004. Liang Chen is the corresponding author.

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

# Appendix

## Table of Contents

## A Broader impact

Our extension of state space models for temporal graph modeling may have broader impacts, particularly if applied to social, traffic, and financial networks which could affect individuals and society. While our work is fundamental and not tied to specific applications, the potential for misuse in surveillance, exacerbation of biases in algorithmic decision-making, or violation of privacy cannot be dismissed. For example, more accurate temporal graph models might inadvertently facilitate more intrusive tracking of individuals or groups, or could be employed in creating discriminatory financial models. It is the responsibility of those employing such technologies to consider these ethical implications and to implement measures such as algorithmic fairness checks, privacy-preserving methodologies, and security protocols that prevent exploitation of the technology. As with any powerful tool, the utmost caution should be exercised to avoid the irresponsible use of our advancements in modeling dynamic systems.

## B Notes

### B.1 Laplacian regularization, diffusion and GNN approximation

In this section, we discuss in detail the smoothness regularization of different types of Laplacians, and their approximations related to popular GNN architectures.

**Inductive bias and compression capability of different Laplacians.** As mentioned in section 3.1, two typical (normalized) graph Laplacians are

$$L_{\text{sym}}(s) = I - D(s)^{-1/2} A(s) D(s)^{-1/2} \tag{11}$$

$$L_{\text{rw}}(s) = I - D(s)^{-1} A(s), \tag{12}$$

with corresponding penalties written as

$$\int_0^t Z(s)^\top L_{\text{sym}}(s) Z(s) d\mu_t(s) = \int_0^t \sum_{(u,v)\in E(s)} \left( \frac{z_u(s)}{\sqrt{d_u(s)}} - \frac{z_v(s)}{\sqrt{d_v(s)}} \right)^2 d\mu_t(s) \qquad (13)$$

$$\int_0^t Z(s)^\top L_{\text{rw}}(s) Z(s) d\mu_t(s) = \int_0^t \sum_{(u,v)\in E(s)} \frac{1}{d_u} \left( z_u(s) - z_v(s) \right)^2 d\mu_t(s). \qquad (14)$$

The above display reveals the inductive bias of Laplacian regularizations as a promoting closeness in a weighted $\ell_2$ metric regarding adjacent nodes' memory compressions, with distinct choices of Laplacians utilizing different weighting schemes. In particular, let $\alpha \to \infty$ in objective 3 then when the Laplacian is chosen as $L_{\text{sym}}$, the solution $Z^{\text{sym}}(s)$ must satisfy

$$\frac{z_v^{\text{sym}}(s)}{\sqrt{d_v(s)}} = \frac{z_u^{\text{sym}}(s)}{\sqrt{d_u(s)}}, \forall (u,v) \in E(s), 0 \le s \le t \qquad (15)$$

It then follows that $Z^{\text{sym}}$ compresses all the historical *degree profiles* over connected components of $G$. Analogously, when $L_{\text{rw}}$ is chosen, it follows that the solution $Z^{\text{rw}}(s)$ must satisfy

$$z_v(s) = z_u(s), \forall (u,v) \in E(s), 0 \le s \le t \qquad (16)$$

which compresses the composition of connected components of $G$.

**Diffusion and GNN approximation.** We consider approximations of the following diffused node features with respect to some type of Laplacian:

$$H = \{h_v\}_{v\in V} := (I + \alpha L)^{-1} X B^\top = \left( I + \sum_{k=1}^\infty (-1)^k \alpha^k L^k \right) X B^\top \qquad (17)$$

The right-hand side of the preceding display is equivalent to performing infinite rounds of message passing. If we drop most of the higher-order terms, we arrive at models similar to graph neural networks. In particular, we keep only the first order terms, i.e., $k = 1$, then for the two Laplacians listed above, for each $v \in V$, we have the resulting approximations:

$$h_v^{\text{sym}} \approx (1-\alpha) B x_v + \sum_{u\in N(v)} \frac{\alpha}{\sqrt{d_u d_v}} B f_u \qquad \text{(GCN-Like)}$$

$$h_v^{\text{rw}} \approx (1-\alpha) B x_v + \sum_{u\in N(v)} \frac{\alpha}{d_u} B f_u. \qquad \text{(SAGE(MEAN)-Like)}$$

The above display exhibits a similar pattern to the design of graph neural networks with a aggregate-then-combine procedure, with the corresponding aggregation steps mirroring two typical GNN architectures GCN [21] and SAGE with mean pooling [16]. Furthermore, note that the effect of the balancing constant $\alpha$ would be absorbed into the learnable parameters of the GNN.

## B.2 An extension to varying node sets

The methodology described in section 3.1 applies to temporal graphs with a *fixed* node set. To extend our approach to accommodate graphs featuring *varying* node sets, we initially focus on the continuous-time context, subsequently delving into discussions on discretization strategies. Suppose on the time interval $\mathcal{T} = [0, T]$, the node set evolves as depicted in the following sequence:

$$V(0) \longrightarrow V(t_1) \longrightarrow V(t_2) \longrightarrow \cdots \longrightarrow V(t_{R-1}) \longrightarrow V(t_R) = V(T). \qquad (18)$$

That is, throughout the interval $\mathcal{T}$, the node set undergoes alterations on $R$ distinct occasions, with associated changes occurring at times $t_1, \ldots, t_R$, respectively. We denote these evolving node sets as $V_0, \ldots, V_R$. To systematically analyze this temporal evolution, we partition the entire interval $\mathcal{T}$ into $R + 1$ segments:

$$\mathcal{T}_r = [t_r, t_{r+1}), 0 \le r \le R \text{ with } t_0 = 0 \text{ and } t_{R+1} = T. \qquad (19)$$

According to the formulations in section 3.1, on each $\mathcal{T}_r$, we have a well defined GHIPPO operator and the solutions are characterized by theorem 1. With an approximation order of $N$, we let the resulting projection coefficients be

$$U_r(t) \in \mathbb{R}^{|V_r| \times N}, 0 \leq r \leq R, t \in \mathcal{T}_r. \tag{20}$$

To address the issue of shape incoherence arising from variations in node sets, we employ a *memory alignment procedure*. This technique facilitates the mapping from $U_r(t_{r+1}-)$ to $U_{r+1}(t_{r+1})$, ensuring that the memory associated with each node is aligned according to the following scheme:

$$u_{v,r+1}(t_{r+1}) = \begin{cases} u_{v,r}(t_{r+1}-) & \text{if } v \in V_r \cap V_{r+1} \\ u_{\text{init}} & \text{if } v \in V_{r+1} \backslash V_r \end{cases}. \tag{21}$$

The memory alignment procedure (21) retains the continuity of states for nodes that persist over time. For nodes that emerge anew within the graph, it assigns a default initial state, which could either be an all-zero state or an estimation derived a priori from the states of neighboring nodes.

**Discretizations.** Within the established context, Theorem 2 remains applicable on each segment $\mathcal{T}_r$. Consequently, our primary concern becomes the treatment of nodes that emerge between consecutive snapshots. Adhering to the ZOH discretization rule, newly emerged nodes lack historical states and therefore do not undergo the MIX strategy, and use their initial state during their first appearance in the recurrent update. This initial state can be set to zero or determined through aggregation from neighboring nodes.

### B.3 Heuristic justifications for layer-specific choice of mixing mechanisms

The various mixing mechanisms introduced in this paper are designed to facilitate an estimation of a weighted average of unobserved graph representations that occur amidst successive observational time points. Starting with the output generated by the initial SSM block, these outputs inherently encapsulate the information pertaining to the current snapshot, as well as that of its antecedent. Thus, the incorporation of mixing mechanisms at a second-layer may inadvertently result in the assimilation of superfluous information, extending beyond the target scope of back-to-back snapshots. Therefore, confining the deployment of mixing solely to the first SSM layer ensures the strict conservation of temporal locality. We have empirically verified that such an approach yields enhanced performances.

## C  Proof of theorems

In this section we present the proof of theorem 1 and theorem 2. We first present some necessary technical preparations: For any $t \in [0, T]$, let $\mu_t$ be some finite measure and let $\mathcal{H}_{\mu_t}$ denote the Hilbert space induced by the inner product

$$\langle f, g \rangle_{\mu_t} := \int_0^t f(s)g(s)d\mu_t(s). \tag{22}$$

Let $\mathcal{P}_N(t)$ be the space of polynomials constructed via the restriction of each element in $\mathcal{P}_N$ to $[0, t]$. We assume the measure family be chosen such that $\mathcal{P}_N(t) \subset \mathcal{H}_{\mu_t}, \forall t \in [0, T]$. For each $v \in V$, we assume that the restriction of $x_v$ (viewing as a function on $[0, T]$) to $[0, t]$ is an element of $\mathcal{H}_{\mu_t}$. Note that these assumptions are trivially satisfied for the scaled Legendre measure (LegS) $\mu_t = \frac{1}{t}\mathbb{I}_{[0,t]}$.

### C.1  Proof of theorem 1

*Proof of theorem 1.* Hereafter we omit the dependence on $\mu_t$ and write the inner product simply as $\langle \cdot, \cdot \rangle$ without misunderstandings. Let $P_0, \ldots, P_{N-1}$ be a set of orthogonal polynomials in $\mathcal{P}_N$ with $\langle P_i, P_j \rangle = 0$ for $i \neq j$ and the degree of $P_n$ is $n$ for each $0 \leq n \leq N - 1$. Then for any $f \in \mathcal{H}_{\mu_t}$, the optimal approximation in $L_2(\mu_t)$ distance in $\mathcal{P}_N$ is given by

$$\Pi(f) = \sum_{n=0}^{N-1} \langle f, P_n \rangle \frac{P_n}{\|P_n\|_{\mu_t}^2}, \tag{23}$$

where we define $\Pi$ to be the projection operator. Now we turn to $\mathcal{L}_t(Z; G, X, \mu)$, viewing $x_v$ as a function on $[0, t]$ for any $v$, we have:

$$\mathcal{L}_t(Z; G, X, \mu) = \int_0^t \sum_{v \in V} (x_v(s) - z_v(s))^2 d\mu_t(s) + \alpha \int_0^t Z(s)^\top L(s) Z(s) d\mu_t(s) \tag{24}$$

$$= \int_0^t \sum_{v \in V} (x_v(s) - \Pi(x_v)(s))^2 d\mu_t(s) \tag{25}$$

$$+ \int_0^t \sum_{v \in V} (\Pi(x_v)(s) - z_v(s))^2 d\mu_t(s) + \alpha \int_0^t Z(s)^\top L(s) Z(s) d\mu_t(s) \tag{26}$$

$$:= \int_0^t \sum_{v \in V} (x_v(s) - \Pi(x_v)(s))^2 d\mu_t(s) + \underline{\mathcal{L}}_t(Z; G, X, \mu) \tag{27}$$

The preceding display suggest that the minimizer of $\mathcal{L}_t(Z; G, X, \mu)$ is the same as the minimizer of $\underline{\mathcal{L}}_t(Z; G, X, \mu)$. It thus suffices to analyze $\underline{\mathcal{L}}_t(Z; G, X, \mu)$ which is easier to work with since $\Pi(x_v) \in \mathcal{P}_N, \forall v \in V$ and the solution is a direct application of Laplacian regularization with respect to the integrand at any $s \in [0, t]$, yielding:

$$\text{GPROJ}_t (G, X) (s) = (1 + \alpha L(s))^{-1} \Pi(X)(s), \tag{28}$$

Now let the coefficient matrix $Q \in \mathbb{R}^{N_V \times N}$ be defined as $Q_{v,n} = \langle x_v, P_n \rangle, \forall v \in V, n \in [N]$, we obtain the GHIPPO operator as:

$$\text{GHIPPO} (G, X) (s) := U(s) = (1 + \alpha L(s))^{-1} Q(s) \tag{29}$$

Next we take derivatives to the coefficients. Note that $L(t)$ is discontinuous and we can only apply derivative on intervals where $L(t)$ remains same. First note that if we choose $\mu_t$ to be the scaled Legendre measure (LegS) with $\mu_t = \frac{1}{t} \mathbb{I}_{[0,t]}$, and $P_n$ as basic Legengre polynomials [11, Appendix B.1.1], then we have the HIPPO property:

$$\frac{dQ(t)}{dt} = Q(t) A^\top + X(t) B^\top \tag{30}$$

where $A \in \mathbb{R}^{N \times N}, B \in \mathbb{R}^{N \times 1}$ with

$$A_{nk} = - \begin{cases} \sqrt{(2n+1)(2k+1)} & \text{if } n > k, \\ n + 1 & \text{if } n = k, \\ 0 & \text{if } n < k, \end{cases} \qquad B_n = \sqrt{2n+1} \tag{31}$$

Fix some $1 \leq m \leq M$ and for $t \in [t_{m-1}, t_m)$ we have:

$$\frac{dU(t)}{dt} = \left( (1 + \alpha L(t))^{-1} \right) \frac{dQ(t)}{dt} \tag{32}$$

$$= (1 + \alpha L(t))^{-1} \left( Q(t) A^\top + X(t) B^\top \right) \tag{33}$$

$$= U(t) A^\top + (1 + \alpha L(t))^{-1} X(t) B^\top. \tag{34}$$

which finishes the proof. $\qquad \square$

## C.2 Proof of theorem 2

*Proof of theorem 2.* For ease of presentation, we operate on the node level instead of graph level. Recall the unobserved dynamics:

$$G_{l-1} = G_{l,0} \xrightarrow{\mathcal{E}_{l,1}} G_{l,1} \xrightarrow{\mathcal{E}_{l,2}} G_{l,2} \longrightarrow \cdots \longrightarrow G_{l,M_l-1} \xrightarrow{\mathcal{E}_{l,M_l}} G_{l,M_l} = G_l \tag{35}$$

Following the assumptions, we can intuitively write the update process as follows:

$$U_{l-1} = U_{l,0} \xrightarrow{G_{l,1}, X_{l,1}} U_{l,1} \xrightarrow{G_{l,2}, X_{l,2}} U_{l,2} \longrightarrow \cdots \longrightarrow U_{l,M_l-1} \xrightarrow{G_{l,M_l}, X_{l,M_l}} U_{l,M_l} = U_l \tag{36}$$

For each $0 \le i \le M_l$, let $D_i := (I + \alpha L_{l,i})^{-1} X_{l,i} B^\top$. Let $d_{v,i}$ be the $v$-th row of $D_i$ and $u_{v,i}$ be the $v$-th row of $U_{l,i}$. We first write the ZOH update corresponding to each step in (36) for every $v \in V$:

$$u_{v,i} = \begin{cases} e^{(t_i - t_{i-1}A)} u_{v,i-1} + A^{-1}\left(e^{(t_i - t_{i-1}A)} - I\right) d_{v,i}, & \text{for } 1 \le i \le M_l \\ u_{v,l} & \text{for } i = 0 \end{cases} \tag{37}$$

Next we do the recursion from the rightmost to the leftmost according to (8):

$$\begin{aligned} u_{v,l} &= e^{(\tau_l - t_{M_l})A} u_{v,M_l} + A^{-1}\left(e^{(\tau_l - t_{M_l})A} - I\right) d_{v,M_l} \\ &= e^{(\tau_l - t_{M_l})A}\left(e^{(t_{M_l} - t_{M_l-1})A} u_{v,M_l-1} + A^{-1}\left(e^{(t_{M_l} - t_{M_l-1})A} - I\right) u_{v,M_l-1}\right) \\ &\quad + A^{-1}\left(e^{(\tau_l - t_{M_l})A} - I\right) u_{v,M_l} \\ &\cdots \\ &= e^{(\tau_l - \tau_{l-1})A} u_{v,l-1} + \Upsilon \end{aligned} \tag{38}$$

where we define

$$\Upsilon = A^{-1}\left(e^{(\tau_l - t_{M_l})A} - I\right) u_{v,M_l} + \sum_{i=1}^{M_l} e^{(\tau_l - t_i)A} A^{-1}\left(e^{(t_i - t_{i-1})A} - I\right) u_{v,i-1} \tag{39}$$

in the above display we define $t_0 = \tau_{l-1}$. Note that $A^{-1}$ and $e^{A\beta}$ are simultaneouly diagonalizable for any $\beta$, therefore the matrix multiplication commutes and we further write

$$\Upsilon = A^{-1}\left(e^{(\tau_l - t_{M_l})A} - I\right) u_{v,M_l} + \sum_{i=1}^{M_l} A^{-1}\left(e^{(\tau_l - t_{i-1})A} - e^{(\tau_l - t_i)A}\right) u_{v,i-1} \tag{40}$$

With some abuse of notation now we let $A \in \mathbb{R}^N$ denote the diagonal vector of the matrix. We provide the following construction:

$$\lambda_i = \begin{cases} \dfrac{e^{(\tau_l - t_{M_l})A} - I}{e^{(\tau_l - \tau_{l-1})A} - I} & i = M_l \\ \dfrac{e^{(\tau_l - t_{i-1})A} - e^{(\tau_l - t_i)A}}{e^{(\tau_l - \tau_{l-1})A} - I} & 0 \le i \le M_l - 1 \end{cases}. \tag{41}$$

Here note that $\lambda_i \in \mathbb{R}^N$. It is straightforward to verify that:

$$\Upsilon = A^{-1}\left(e^{(\tau_l - \tau_{l-1})A} - I\right) \sum_{i=0}^{M_l} \lambda_i \odot u_{v,i} \tag{42}$$

where $\{\lambda_i\}_{0 \le i \le M_l}$ are non-negative $N$-dimensional vectors satisfying $\sum_{i=0}^{M_l} \lambda_i = \mathbf{1}_N$, with $\mathbf{1}_N$ denoting the all-one vector of dimension $N$. As the values of $\lambda$s are *independent* of $v$, the proof finishes by combining (38), (42) and write the above conclusion in matrix form via setting $\Lambda_i = \text{diag}(\lambda_i), 0 \le i \le M_l$ $\qquad\square$

# D  Algorithm descriptions

## D.1  The designs of mixing mechanism MIX

We consider two types of mixing mechanisms: convolution (CONV1D) and Scaled interpolation (INTERP) which we describe below:

**CONV1D.**  This is the usual convolution operation along the sequence dimension using *shared parameters*. We use a kernel size of 2 so that only consecutive representations are mixed.

**INTERP.** This is an input-dependent weighted average strategy followed by an input-dependent scaling, implemented as

$$\text{MIX}\,(Z_1, Z_2) = \rho(Z_1, Z_2) \odot (\xi(Z_1, Z_2) \odot Z_1 + (1 - \xi(Z_1, Z_2)) \odot Z_2)\,, \tag{43}$$

where $Z_1, Z_2 \in \mathbb{R}^{N_V \times d}$ are node representation matrices corresponding to consecutive snapshots. $\rho$ and $\xi$ are scale and weight functions that map two inputs into positive real numbers of identical shape with $Z_1$ or $Z_2$, defined by

$$\rho(Z_1, Z_2) = \text{softplus}\,(W_\rho[Z_1\|Z_2] + b_\rho)\,, \quad \xi(Z_1, Z_2) = \text{sigmoid}\,(W_\xi[Z_1\|Z_2] + b_\xi) \tag{44}$$

where $W_\rho, W_\xi \in \mathbb{R}^{2d \times d}$ and $b_\rho, b_\xi \in \mathbb{R}^d$ are learnable parameters therein.

## D.2 Details of GRAPHSSM

In this section, we elucidate on the methodology of GRAPHSSM through three specific instantiations. For clarity in our explanation, we employ certain notational conventions that might be somewhat different from the main text: the term $V$ refers to the number of vertices in each graph snapshot $G_l$ within a sequence of $L$ graph snapshots $\{G_l\}_{1 \leq l \leq L}$ which we further denote as $G_{1:L}$, and $D$ represents the dimensionality of node features. The symbol LINEAR is used to represent a linear projection layer including a bias term, where the dimensions for input and output are typically clear from the context to ensure compatibility. The notation $X_{1:L}$ denotes the concatenation of $L$ tensors of the same dimensions along their second axis. For operations on tensors of order higher than two, we use the einsum notation, as defined by the einops framework [37]. We present the algorithmic description of our design of SSM layers, namely GRAPHSSM-S4 (resp. GRAPHSSM-S5, GRAPHSSM-S6) in algorithm 1 (resp. algorithm 2, algorithm 3). [7] Subsequently, we adopt the

---

**Algorithm 1** GRAPHSSM-S4 layer

---

**Input:** A sequence of graph (snapshots) $G_{1:L}$ with each of size $V$.
    Node (hidden) feature inputs $X_{1:L} \in \mathbb{R}^{V \times L \times D}$.
    A graph neural network $\text{GNN}_\theta$ parameterized by $\theta$.
    A mixing mechanism $\text{MIX}_\phi$ parameterized by $\phi$.
    State-space parameters $A \in \mathbb{R}^{D \times N}, B \in \mathbb{R}^{D \times N}, C \in \mathbb{R}^{D \times N}$.
    A linear layer for adaptive time gaps $\text{LINEAR}_\tau$.
**Output:** $Y_{1:L} \in \mathbb{R}^{V \times L \times D}$

1: # Approximate diffusion via GNN
2: **for** $t = 1$ to $L$ **do**
3:     $Z_l = \text{GNN}_\theta(X_l, G_l)$;
4:     $H_l = Z_l$ if $l = 1$ else $\text{MIX}(Z_l, Z_{l-1})$;
5: **end for**
6: Initialize state $U_0 = 0$; # SISO state of shape $V \times D \times N$
7: **for** $t = 1$ to $L$ **do**
8:     $\Delta_l = \text{softplus}\,(\text{LINEAR}_\tau(H_l))$;
9:     $\overline{A} = \exp\,(\text{einsum}(\Delta_l, A, "V, DN \rightarrow VDN"))$;
10:     $\overline{B} = \text{einsum}(\Delta_l, B, "V, DN \rightarrow VDN")$;
11:     $U_l = U_{l-1} \odot \overline{A} + \text{einsum}(\overline{B}, H_l, "VDN, VD \rightarrow VDN")$;
12:     $Y_l = \text{einsum}(U_l, C, "VDN, DN \rightarrow VD")$;
13: **end for**;
14: **return** $Y_{1:L}$;

---

following neural architecture composed of $K$ blocks, with each block composed of one SSM layer followed by nonlinear activation and a residual connection:

$$H_{1:L}^{(k)} = \sigma\left(\text{SSMLAYER}\left(H_{1:L}^{(k-1)}, G_{1:L}\right)\right) + \text{LINEAR}\left(H_{1:L}^{(k-1)}\right), 1 \leq k \leq K, \tag{45}$$

where $H_{1:L}^{(0)}$ are the node features $X_{1:L}$. The SSMLAYER in (45) may be chosen as any of $\{\text{GRAPHSSM-S4}, \text{GRAPHSSM-S5}, \text{GRAPHSSM-S6}\}$. In our implementation of GRAPHSSM-S6, we add an additional layer normalization as the last operation of each block.

---

[7]In these algorithmic descriptions, we illustrate using the representation mixing mechanism. The case for feature mixing is similarly defined.

---

**Algorithm 2** GRAPHSSM-S5 layer

---

**Input:** A sequence of graph (snapshots) $G_{1:L}$ with each of size $V$.
     Node (hidden) feature inputs $X_{1:L} \in \mathbb{R}^{V \times L \times D}$.
     A graph neural network $\text{GNN}_\theta$ parameterized by $\theta$.
     A mixing mechanism $\text{MIX}_\phi$ parameterized by $\phi$.
     State-space parameters $A \in \mathbb{R}^{N \times 1}, B \in \mathbb{R}^{D \times N}, C \in \mathbb{R}^{N \times D}$.
     A linear layer for adaptive time gaps $\text{LINEAR}_\tau$.
**Output:** $Y_{1:L} \in \mathbb{R}^{V \times L \times D}$

  1: # Approximate diffusion via GNN
  2: **for** $t = 1$ to $L$ **do**
  3:      $Z_l = \text{GNN}_\theta(X_l, G_l)$;
  4:      $H_l = Z_l$ if $l = 1$ else $\text{MIX}(Z_l, Z_{l-1})$;
  5: **end for**
  6: Initialize state $U_0 = 0$; # MIMO state of shape $V \times N$
  7: **for** $t = 1$ to $L$ **do**
  8:      $\Delta_l = \mathsf{softplus}\left(\text{LINEAR}_\tau(H_l)\right)$;
  9:      $\overline{A} = \exp\left(\Delta_l A^\top\right)$;
10:      $\overline{B} = \mathsf{einsum}(\Delta_l, B, "V, DN \to VDN")$;
11:      $U_l = U_{l-1} \odot \overline{A} + \mathsf{einsum}(\overline{B}, H_l, "VDN, VD \to VN")$;
12:      $Y_l = U_l C$;
13: **end for**;
14: **return** $Y_{1:L}$;

---

**Initialization strategy.** Recent developments in state space modeling have underscored the significance of initializing the state matrices $A$, $B$, and $C$, with the initialization of $A$ frequently emerging as the most critical factor for the performance of the SSM [12]. Building upon the progress made in S4 [13] and S4D [29, 15], we evaluate three disparate initialization strategies for the matrix $A$. Note that since $A$ is diagonal, we instead represent $A$ as a $N$-dimensional vector:

$$\forall 1 \leq n \leq N : \quad A_n^{\text{S4D-Real}} = -(n+1), \quad A_n^{\text{S4D-Const}} \equiv \frac{1}{2}, \quad A_n^{\text{random}} = -e^\chi \tag{46}$$

**S4D-Real (HIPPO)** This is the diagonal part of the original HIPPO matrices (6).

**S4D-Const (Constant)** This is the real part of the eigenvalues corresponding to the S4N matrix as defined in [13], which equals $-\frac{1}{2}$.

**Random** This initialization is generated via a negative transform of a random number $\chi$, which we generated using the Glorot initialization method.

Additionally, we initialize the $B$ matrices using a constant of all-1 vector, and we initialize $C$ randomly using Glorot.

## D.3 Complexity and implementations

As detailed in section 3.1 and the algorithmic outlines provided, the implementations of GRAPHSSM across all three variants can be stratified into two primary phases: a diffuse-and-mixing step, and a linear recurrence step. The diffuse-and-mixing stage facilitates straightforward parallelization through the employment of methods such as graph batching. The inherent linear characteristic of the recurrence operation permits the utilization of efficient computation strategies, notably the selective scan technique as introduced in [10]. This approach yields a FLOP complexity of $O(VLDN)$ per SSM layer with work-efficient parallelization, concurrently achieving IO efficiency. Furthermore, note that if we replace the adaptive time gap mechanism into a constant, i.e., we use $\Delta_l \equiv \frac{1}{L}, 1 \leq l \leq L$ in line 8 of algorithm 1 and algorithm 2, the resulting linear system is time-invariant and we can use other computational accelerations like convolution [13, 7] and parallel scan [40].

**Algorithm 3** GRAPHSSM-S6 layer
***
**Input:** A sequence of graph (snapshots) $G_{1:L}$ with each of size $V$.
    Node (hidden) feature inputs $X_{1:L} \in \mathbb{R}^{V \times L \times D}$.
    A graph neural network $\text{GNN}_\theta$ parameterized by $\theta$.
    Three graph neural networks for selective state spaces $\text{GNN}_{\theta_B}, \text{GNN}_{\theta_C}, \text{GNN}_\Delta$.
    A mixing mechanism $\text{MIX}_\phi$ parameterized by $\phi$.
    State-space parameters $A \in \mathbb{R}^{D \times N}$.

**Output:** $Y_{1:L} \in \mathbb{R}^{V \times L \times D}$
1: # Approximate diffusion via GNN
2: **for** $t = 1$ to $L$ **do**
3:     $Z_l = \text{GNN}_\theta(X_l, G_l)$;
4:     $H_l = Z_l$ if $l = 1$ else $\text{MIX}(Z_l, Z_{l-1})$;
5: **end for**
6: Initialize state $U_0 = 0$; # SISO state of shape $V \times D \times N$
7: **for** $t = 1$ to $L$ **do**
8:     $\Delta_l = \mathsf{softplus}\,(\text{GNN}_\Delta(X_l, G_l) + b)$;
9:     $\overline{A} = \exp\,(\mathsf{einsum}(\Delta_l, A, "VD, DN \to VDN"))$;
10:     $\overline{B} = \mathsf{einsum}(\Delta_l, \text{GNN}_{\theta_B}(X_l, G_l), "VD, VN \to VDN")$;
11:     $U_l = U_{l-1} \odot \overline{A} + \mathsf{einsum}(\overline{B}, H_l, "VDN, VD \to VDN")$;
12:     $\overline{C} = \text{GNN}_{\theta_C}(X_l, G_l)$;
13:     $Y_l = \mathsf{einsum}(U_l, \overline{C}, "VDN, DN \to VD")$;
14: **end for**;
15: **return** $Y_{1:L}$;
***

# E    Discussions and limitations

In this section, we discuss the limitations of the GRAPHSSM framework and propose a few future research directions that might be of interest.

## E.1    Extension to continuous-time temporal graphs

In this study, we focus on modeling discrete-time temporal graphs (DTTGs) through the lens of discretizing continuously evolving systems. The continuous-time viewpoint holds promise for encapsulating the modeling of continuous-time temporal graphs (CTTGs), a domain of growing importance in graph learning literature. However, the current GHiPPO framework has its limitations when extending to continuous-time setups. We provide a brief discussion as follows:

**DTDG, CTDG and GHiPPO**    Recall that in our formulation of the underlying graph process (2), the node features evolve continuously and the topological relations among nodes allow finite (countable) mutations. In DTTG representations, we do not directly observe the events, but we observe the entire graph at certain time spots resulting in a serious of snapshots. In this spirit, DTTGs have complete *latitudinal* information, but are lossy regarding *longitudinal* information. In CTTG representations, we have complete observations of events, but upon each event information, we do not observe the features of the rest of the nodes (that do not participate in those specific events). Therefore, CTTGs have complete longitudinal information, but are lossy regarding latitudinal information. In this regard, we may view DTDG and CTDG as two different lossy observation schemes of the underlying graph process in the GHiPPO abstraction.

**Handing CTTGs using SSM discretizations is challenging**    In section 3.2 of our paper (especially theorem 2), we established the discretization scheme upon an ideal, discrete observation (We observe the graph snapshot at each mutation events). We believe that this result might reasonably hints the gap between possible empirical approximations in either DTTG or CTTG scenarios: In DTTGs, we believe approximations using available snapshots are possible since from hindsight, the ideal representation is a convex combination of the snapshot representations at the mutation times. The approximation bias mostly comes from fewer snapshots, and we use mixing strategies to mitigate

the biases. However, in CTTG scenarios, we miss the majority of information in each snapshot. Besides, consturcting snapshots from CTDGs is itself a very impractical method. Hence, we regard the modeling of CTDG to be beyond the scope of GraphSSM.

## E.2  Going beyond piecewise dynamics

The distinguishing algorithmic feature of GHIPPO compared to the conventional HIPPO framework lies in the piecewise nature of the dynamical system it generates. This characteristic leads to the challenge of dealing with unobserved dynamics, a factor that motivated the development of our MIX module. However, it's important to acknowledge that the mixing module serves as an approximation of the actual underlying dynamics, thus representing a limitation within the framework. This acknowledgment raises an intriguing question: might there exist alternative problem formulations capable of yielding a smoother dynamical system that mitigates the issue of discontinuities? One potential pathway could involve adopting smoother versions of the Laplacian or revising the approximation objective specified in (3) towards one that fosters a smooth solution. Such a solution would promote consistency in the dynamics across the complete temporal interval. Implementing these innovations would, however, necessitate the incorporation of more sophisticated technical assumptions and theoretical tools which we left for future explorations.

## F  Experimental setup

Table 6: Dataset Statistics.

|  | DBLP-3 | Brain | Reddit | DBLP-10 | arXiv | Tmall |
|---|---|---|---|---|---|---|
| **#Nodes** | 4,257 | 5,000 | 8,291 | 28,085 | 169,343 | 577,314 |
| **#Edges** | 23,540 | 1,955,488 | 264,050 | 236,894 | 2,315,598 | 4,807,545 |
| **#Features** | 100 | 20 | 20 | 128 | 128 | 128 |
| **#Classes** | 3 | 10 | 4 | 10 | 40 | 5 |
| **#Time Steps** | 10 | 12 | 10 | 27 | 35 | 186 |
| **Category** | Citation | Biology | Society | Citation | Citation | E-commerce |
| **$\text{TC}_{\text{structure}}$** | 0.139 | 0.024 | 0.030 | 0.823 | 0.580 | 0.811 |
| **$\text{TC}_{\text{feature}}$** | 0.468 | 0.070 | 0.556 | 0.823 | 1.000 | 0.712 |

**Temporal continuity.** As illustrated in figure 2, our work has highlighted the problem of unobserved graph mutations in learning from discrete-time temporal graphs. The issue of unobserved graph mutations greatly hampers the temporal continuity of such graphs, presenting a significant challenge for learning if not properly addressed. To quantitatively measure the temporal continuity of a temporal graph, we calculate the average proximity between consecutive graph snapshots in the graph sequence. Specifically, we utilize Jaccard distance and Cosine similarity to measure the temporal continuity in terms of graph structure and node features, respectively:

$$\text{TC}_{\text{structure}} = \frac{1}{L-1} \sum_{l}^{L-1} \frac{\mathcal{E}_l \cap \mathcal{E}_{l+1}}{\mathcal{E}_l \cup \mathcal{E}_{l+1}},$$

$$\text{TC}_{\text{feature}} = \frac{1}{L-1} \sum_{l}^{L-1} \text{Sim}(X_l, X_{l+1}), \tag{47}$$

$$\text{where} \quad \text{Sim}(X_l, X_{l+1}) = \frac{1}{N_V} \sum_{v \in V} \frac{\langle x_{l,v}, x_{l+1,v} \rangle}{\|x_{l,v}\| \, \|x_{l+1,v}\|}.$$

**Datasets.** We focus on the node classification task in discrete-time temporal graphs, which is a straightforward extension of static graphs. The experiments are conducted on six temporal graph benchmarks with different scales and time snapshots, including DBLP-3 [47], Brain [47], Reddit [47], DBLP-10 [25], arXiv [19], and Tmall [25]. Dataset statistics are summarized in table 6 including the corresponding temporal continuity. The graph datasets are collected from real-world networks belonging to different domains. It should be noted that in the arXiv dataset, the time information is associated with the nodes rather than the edges. As a result, we split the snapshots of arXiv based on

the occurrence of nodes. Each snapshot graph in the dataset shares the same attribute information but not the topology. Therefore, $TC_{feature} = 1.000$ for arXiv in our experiments.

**Baselines.** We compare GRAPHSSM with the following baselines: (i) static graph embedding methods: DeepWalk [36], Node2Vec [9]; (ii) temporal graph embedding methods: HTNE, M$^2$DNE, and DynamicTriad [50]; (iii) discrete-time temporal graph neural networks: MPNN [32], STAR [47], tNodeEmbed [39], EvolveGCN [34], SpikeNet [25], and ROLAND [48]. For baselines that are originally designed for static graphs, we accumulate historical information (edges) in the graph snapshot sequence and represent the static graph structure at the last time point. All baselines are carefully tuned to achieve their best results based on the code officially provided by the authors.

**Implementation details.** GRAPHSSM is built on the success of SSMs, where in this work we have derived variants of GRAPHSSM-S4, GRAPHSSM-S5, and GRAPHSSM-S6, under different SSM settings. Our experiments are mainly conducted on the S4 architecture. we employ feature mixing for DBLP-10 and representation mixing for other datasets. The graph convolution networks used to learn the graph structure are SAGE [16] for all datasets, except for arXiv, where GCN [21] is used. We implement our models as well as baselines with PyTorch [35] and PyTorch Geometric [5], which are open-source software released under BSD-style [8] and MIT [9] license, respectively. All datasets used throughout experiments are publicly available. All experiments are conducted on an NVIDIA RTX 3090 Ti GPU with 24 GB memory. Code will be made available at `https://github.com/EdisonLeeeee/GraphSSM`.

**Evaluation protocol.** We adopt the conventional *transductive* learning setting, where the graph structure of all snapshots is visible during both training and inference stages. This is analogous to the standard node classification task, but with the additional incorporation of time information to facilitate the learning. For the DBLP-3, Brain, and Reddit datasets, we adopt the 81%/9%/10% train/validation/test splits as suggested in [47]. For the DBLP-10 and Tmall datasets, we follow the experimental settings of previous works [25], where 80% of the nodes are randomly selected as the training set, and the remaining nodes are used as the test set. Note that stratified sampling is employed to ensure that the class distribution remains consistent across splits. For the arXiv dataset, we use the fixed public splits. We use Micro-F1 and Macro-F1 to evaluate the node classification performance. We report the average performance with standard deviation across 5 runs for each method.

---

[8] `https://github.com/pytorch/pytorch/blob/master/LICENSE`
[9] `https://github.com/pyg-team/pytorch_geometric/blob/master/LICENSE`

