# OpenReview forum: "State Space Models on Temporal Graphs: A First-Principles Study"
_NeurIPS.cc/2024/Conference — NeurIPS 2024 poster_

### Official Review · Reviewer_ZL4u · 2024-07-08

**Soundness:** 2
**Presentation:** 2
**Contribution:** 2
**Rating:** 6
**Confidence:** 3

**Summary:**

The paper proposes an approach for processing discrete-time temporal graphs via an extension of state-space models (SSMs). In this direction, the main contribution of the paper is a generalization of the HIPPO framework for graph structured data (named GHIPPO), which defines how the state of different nodes should be updated through time to compress the evolution of nodes features (a solution capable of dealing with multiple changes that might happen between two snapshots is additionally introduced in the paper). In experimental evaluation, an SSM-based solution implemented with the proposed GHIPPO framework (named GraphSSM) outperforms a variety of approaches on both the moderately sized DBLP-3, Brain, Reddit and DBLP-10 datasets, and the large-scale arXiv and Tmall datasets.

**Strengths:**

While I’m personally not an expert in SSM models, I generally found the paper to be an interesting read (to the best of my knowledge there is not much work on applications of SSMs to graph-structured data, and as such the work highlights a research direction possibly worthy of further exploration). The presentation of the paper is generally understandable (albeit not always clear in some spots, see weaknesses), and allows a reader unfamiliar with the SSM literature to grasp the overall idea of the paper. The experimental evaluation of the paper appears well done (with the proposed approach outperforming prior art on a variety of different datasets), although there might be room for improvement.

**Weaknesses:**

Generally I do not have many weaknesses to highlight in the paper. From a technical perspective, I would want to highlight that:

1) the proposed approach might face some issues with heterophilic graphs (as the smoothness in the node state imposed in equation (3) is not necessarily a good regularizer in that setting - see point 5 below)

2) the proposed methodology is limited to discrete-time temporal graphs, and further work would need to be done to extend the model to a continuous-time setting (this is also highlighted by the authors themselves in the paper).

From a different perspective, a weakness of the paper is probably its exposition, which in some parts is a little confusing and could be improved. In this direction:

1) lines 67, I believe there is a typo and the authors should replace “features the dynamics” with “the features dynamics”
2) line 74 to 77 are quite unclear and should probably be refactored
3) line 105, “recurrence model” should probably be replaced with “recurrent model”
4) Table 1, while it is true that classic transformers have a quadratic space and time complexity, there are also efficient transformers (e.g. Performs) that show linear space and time complexity. I believe these are not mentioned in the paper and probably they should, in order to provide to the reader a complete picture of where we stand with transformer architectures.
5) I believe the smoothness in node state imposed in equation 3 with the dirichlet energy regularizer makes sense only on homophilic graphs, and might not be a meaningful prior to use when dealing with heterophily. This is not pointed out in the paper and probably it should be highlighted as a possible limitation.
6) Lines 180-181, the authors mention that what minimizes equation 3 is a piecewise vector. I believe what they mean is that the solution is piecewise over time, I think this should probably be clarified in the paper as it might be a source of confusion
7) line 184-185, how the nodes' memories are parameterized as polynomials is very unclear here. How are those polynomials defined? What is the input to such polynomials? It might be that a reader familiar with the HIPPO framework would be aware of such details, but the lack of an explanation in the text makes it hard to draw a complete picture here.
8) lines 335-336, GraphSSM with S5 doesn’t always achieve poor performance (it is indeed the best performing model on DBLP-3 in table 4). This should be clarified in the text.
9) line 339, I believe there is a typo and the “and” at the beginning of the sentence should be fixed.

**Questions:**

One thing that I think would be helpful to improve the strength of the paper is to run some experiments where the proposed SSM module is swapped with a RNN or transformer architecture. As we can see from the ablation study of table 5, it appears that the mixing strategy one might decide to use is relevant to outperform prior art. As a result of this, I wonder what would be the performance that a comparable solution would have with the very same architecture used by the authors but without using any SSM approach. This will clearly provide an indication on whether using SSMs is beneficial for achieving the desired performance (which is the main claim of the paper) or whether the results we see in Table 2 and 3 are only due to the mixing strategy.

**Limitations:**

Yes

---

> ### Author Rebuttal · Authors · 2024-08-07
>
> We appreciate your valuable and insightful comments, and we provide our responses below.
>
> **W1: heterophilic graphs issues**
>
> Thank you for your question. We agree with you that equation (3) reflects a prior belief that the node representation generated by GHiPPO should be homophilic under the representation level [3] at any time point. While being empirically effective, it poses significant challenge when the underlying temporal graph has more complex characteristics, and we will discuss it in revisions of our paper. Indeed, we do think that defining a proper homophily metric when the underlying graph is treated as a process is itself a valuable future research question.
>
> **W2: Typos or misunderstandings  in Line 67, Lines 74-77, Line 105, Lines 180-181, and Line 339**
>
> We appreciate the reviewer for pointing out the careless errors. In the revision, we will make thorough proofreading to enhance readability by correcting any typos and improving the writing.
>
> **W3: Table 1 may not be entirely accurate**
>
> Thank you for bringing this to our attention. We agree that Table 1 may not be entirely accurate in all cases. In this context, "Transformers" refers to traditional "softmax Transformers." We will include a note to clarify this and prevent any potential misunderstandings. We will also provide a comprehensive discussion on this matter, as outlined below:
>
> Indeed, there are close relationships between transformers with finite kernel feature maps (which are essentially what efficient transformers rely on), RNNs and SSMs [1, 2]. All of the three allow a recurrent parameterization and constant-memory-linear-compute inference. The primary difference is the update rule during recurrence: Linear transformers essentially use a unitary state matrix that is not learnable, while standard RNNs use learnable but not well-controlled state matrix. SSMs use an improved and learnable state matrix by adopting more careful initializations.
>
> **W4: On semantics of node memory**
>
> We will add a more accessible and detailed introduction to the background on HiPPO abstraction in revisions of our paper. Specifically, by an $N$-dimensional node memory at time $t$ we mean the coefficients of some order-$N$ polynomial (under the orthogonal polynomial basis) that optimally approximate the node features (viewed as a function over time) up until time $t$. The rationale of this representation is that a finite dimensional function space spaned by polynomials are characterized by their coefficients, thereby allowing us to **embed functions into finite dimensional vectors**. Now we answer your question: The inputs to the approximation polynomial is time $t$. The coefficients $u(t)$ of the polynomial is determined by an optimal approximation criteria, and constitues the node memory at time $t$.
>
> **W5: Performance of GraphSSM-S5**
>
> Thank you for your suggestion. We will address it and provide clarification in the revisions.
>
> **W6: Ablation study on swapping SSMs with RNN and Transformer**
>
> Thak you for your insightful suggestion. We replaced the SSM architecture with RNN and Transformer and present the results below. As observed, SSMs prove to be a superior choice compared to RNNs and Transformers. While Transformers excel in NLP tasks, they are not well-suited for discrete graph sequences.
>
> ||**DBLP-3**||**Brain**||**Reddit**||**DBLP-10**||
> |--------------------|:-----------:|:-----------:|:-----------:|:-----------:|:-----------:|:-----------:|:-----------:|:-----------:|
> ||**Micro-F1**|**Macro-F1**|**Micro-F1**|**Macro-F1**|**Micro-F1**|**Macro-F1**|**Micro-F1**|**Macro-F1**|
> |GraphSSM-RNN|84.24±1.3|83.10±1.4|92.28±1.5|92.43±1.0|43.11±1.7|43.15±1.7|73.46±1.5|72.93±1.5|
> |GraphSSM-Transformer|85.02±1.1|84.98±0.8|93.47±1.5|93.11±1.1|43.48±0.7|43.11±0.5|75.65±0.8|74.32±0.6|
> |GraphSSM-S4|85.26±0.9|85.00±1.3|93.52±1.0|93.54±0.9|**49.21±0.5**|**49.05±0.7**|**76.80±0.3**|**76.00±0.4**|
> |GraphSSM-S5|**86.29±1.0**|**85.78±0.9**|93.00±0.4|93.01±0.4|44.75±0.4|44.79±0.4|75.19±0.6|73.95±0.4|
> |GraphSSM-S6|86.10±0.5|85.70±0.6|**93.80±0.3**|**94.47±0.6**|43.11±0.9|42.85±1.1|74.09±0.3|73.16±0.2|
>
> **Q1: Link prediction results**
>
> Thank you for your suggestion. The results for the link prediction task are presented below. We follow the experimental settings of ROLAND, where the model leverages information accumulated up to time t to predict edges in snapshot t + 1.
>
> ||DBLP-3||Brain||Reddit||DBLP-10||
> |-----------|--------------|--------------|--------------|--------------|--------------|--------------|--------------|--------------|
> ||AUC|AP|AUC|AP|AUC|AP|AUC|AP|
> |STAR|94.21±0.43|91.80±0.51|56.10±0.48|55.24±0.71|98.21±0.59|98.32±0.32|86.81±0.23|85.3±0.31|
> |tNodeEmbed|93.19±0.37|90.22±0.42|55.21±0.29|56.32±0.36|98.39±0.41|98.10±0.23|87.32±0.41|86.54±0.25|
> |EvolveGCN|94.01±0.62|91.05±0.39|56.33±0.41|56.91±0.55|98.77±0.39|98.80±0.13|88.32±0.24|87.19±0.50|
> |SpikeNet|92.53±0.57|90.11±0.51|54.95±0.58|55.88±0.64|97.97±0.12|97.06±0.24|86.88±0.17|85.40±0.18|
> |ROLAND|95.01±0.55|91.25±0.38|56.87±0.41|56.02±0.23|98.76±0.11|98.99±0.14|89.42±0.30|88.91±0.23|
> |GraphSSM-S4|95.47±0.23|91.58±0.24|57.74±0.45|56.92±0.21|99.59±0.32|99.24±0.20|90.62±0.45|90.12±0.37|
> |GraphSSM-S5|**96.45±0.15**|92.41±0.17|**58.49±0.33**|**57.40±0.31**|99.66±0.21|99.35±0.12|90.99±0.29|90.80±0.56|
> |GraphSSM-S6|95.88±0.31|**92.52±0.28**|56.77±0.41|57.16±0.25|**99.70±0.13**|**99.42±0.14**|**91.16±0.32**|**91.19±0.24**|
>
>
> [1] Transformers are rnns: Fast autoregressive transformers with linear attention. ICML, 2020.
> [2] Transformers are SSMs: Generalized models and efficient algorithms through structured state space duality. arXiv 2024.
> [3] Luan, Sitao, et al. "Is heterophily a real nightmare for graph neural networks to do node classification?." arXiv preprint arXiv:2109.05641 (2021).
>
> ---
>
> We hope our responses were helpful in adequately addressing your earlier concerns. In case you have any further questions or comments, please let us know, and we will gladly respond.

---

> > ### Comment · Reviewer_ZL4u · 2024-08-08
> >
> > I would like to thank the authors for their response. I don't have further question and I'm happy to raise my score to a 6

---

> ### Author Response · Authors · 2024-08-08
>
> Thank you for your response and for raising the score. We are pleased to address the concerns you have raised.

---

### Official Review · Reviewer_PUCW · 2024-07-09

**Soundness:** 3
**Presentation:** 3
**Contribution:** 3
**Rating:** 5
**Confidence:** 3

**Summary:**

The paper introduces GRAPHSSM, a novel state space model framework for temporal graphs. GRAPHSSM extends state space models (SSMs) by incorporating structural information through Laplacian regularization to handle the dynamic behaviors of temporal graphs. The framework aims to overcome limitations of recurrent neural networks (RNNs) and Transformers in modeling long-range dependencies and managing computational complexity. The authors propose the GHIPPO abstraction for memory compression and various mixing mechanisms to handle unobserved graph mutations. Extensive experiments on multiple temporal graph benchmarks demonstrate the effectiveness and efficiency of GRAPHSSM.

**Strengths:**

- The paper is well-written and I enjoy reading the paper. The derivation of the method and the extension from SSM is technically sound

- The the overall research direction is well-motivated and interesting. SSM is indeed a good potential candidate for modelling temporal graph.

**Weaknesses:**

- There already exists an important family of temporal graph neural networks (TGNNs) that use memory mechanism (see [1] and [2] for examples) and resemble a state transition model. The paper seems to neglect/be unaware of this line of related work.

- There are parts of the method/extension that are not clear (see questions below for more details)

- There are a couple of serious limitations regarding the empirical study:
1) the baselines for the paper are rather outdated. The paper did not compare with the more recent TGNN models (e.g. above-mentioned memory-based temporal graph neural network, which is the SOTA for TGNN)
2) the performance improvement from the proposed baselines is still marginal
3) the conical task for temporal graph neural should be link prediction which is missing from the current study

[1] Rossi, Emanuele, et al. "Temporal graph networks for deep learning on dynamic graphs." arXiv preprint arXiv:2006.10637 (2020).

[2] Zhang, Yao, et al. "Tiger: Temporal interaction graph embedding with restarts." Proceedings of the ACM Web Conference 2023. 2023.

**Questions:**

- Q1: I fail to understand the second part of Eq.(3).  Why does minimizing the second term of Eq.(3) amounts to increasing smoothness?  In addition, I don't think the interpretation/remark given in line 179-180 are correct. If $\alpha$ goes to infinity, the whole objective, Eq.(3), is dominated by the second term. There is a degenerate solution $Z(s) = \mathbf{0}$ (zero vector) that would minimize the objective. Which part of the objective/model prevent this degenerate solution?

- Q2: Theorem 1 is saying that GHIPPO is "nice" because its parameter update can be described by the given ODEs. Can you further explain why this property is nice? does the benefit comes from better efficiency or inference performance? and how?

- Q3: the discretization in the HIPPO paper was for computation purposes. In the case of temporal graph, where event and snapshot are inherently discrete. What is the (optimization) objective for discretization? i.e., what should be considered as a "good discretization" in this case?

**Limitations:**

N.A

---

> ### Author Rebuttal · Authors · 2024-08-07
>
> **W1: Missing comparison of related works [1,2] and advanced methods**
>
> Thank you for your suggestions. We have throughly reviewed the literature in temporal graph learning and have awared of several advanced works such as [3] and [4]. However, to our best knowledge, they are focusing on the **continuous-time tempora graph leanring** research and it is unfair to consider them as baselines for comparison under the discrete-time settings. We have explicitly clarified the situation in Related Work, and we will add a brief discussion with GraphSSM and continus-time methods in the next revision.
>
>
> **W2: Performance improvement is marginal**
>
> First of all, we respectively disagree with that the performance improvement is marginal. GraphSSM consistently achieves state-of-the-art performance and outperforms the runner-up baselines by significant margins, especially on the Reddit dataset. Secondly, scientific research is not a race; it is essential to place strong emphasis on other aspects of a method besides a single performance metric. GraphSSM shows superior performance while also exhibiting lower memory and computational overheads compared to other methods. This is also an important contribution of our work.
>
> **W3: Link prediction results**
>
> Thank you for your suggestion. The results for the link prediction task are presented below. We follow the experimental settings of ROLAND, where the model leverages information accumulated up to time t to predict edges in snapshot t + 1.
>
> ||DBLP-3||Brain||Reddit||DBLP-10||
> |-----------|--------------|--------------|--------------|--------------|--------------|--------------|--------------|--------------|
> ||AUC|AP|AUC|AP|AUC|AP|AUC|AP|
> |STAR|94.21±0.43|91.80±0.51|56.10±0.48|55.24±0.71|98.21±0.59|98.32±0.32|86.81±0.23|85.3±0.31|
> |tNodeEmbed|93.19±0.37|90.22±0.42|55.21±0.29|56.32±0.36|98.39±0.41|98.10±0.23|87.32±0.41|86.54±0.25|
> |EvolveGCN|94.01±0.62|91.05±0.39|56.33±0.41|56.91±0.55|98.77±0.39|98.80±0.13|88.32±0.24|87.19±0.50|
> |SpikeNet|92.53±0.57|90.11±0.51|54.95±0.58|55.88±0.64|97.97±0.12|97.06±0.24|86.88±0.17|85.40±0.18|
> |ROLAND|95.01±0.55|91.25±0.38|56.87±0.41|56.02±0.23|98.76±0.11|98.99±0.14|89.42±0.30|88.91±0.23|
> |GraphSSM-S4|95.47±0.23|91.58±0.24|57.74±0.45|56.92±0.21|99.59±0.32|99.24±0.20|90.62±0.45|90.12±0.37|
> |GraphSSM-S5|**96.45±0.15**|92.41±0.17|**58.49±0.33**|**57.40±0.31**|99.66±0.21|99.35±0.12|90.99±0.29|90.80±0.56|
> |GraphSSM-S6|95.88±0.31|**92.52±0.28**|56.77±0.41|57.16±0.25|**99.70±0.13**|**99.42±0.14**|**91.16±0.32**|**91.19±0.24**|
>
>
> **Q1: On remark 1 and degenerate solutions**
>
> Thank you for pointing out the degenerate solution and we shall restate the remark in future revisions of our paper. Technically, the minimizers of Laplacian quadratic forms shall satisfy a smoothness condition over connected components of the underlying graph (please refer to our discussion in appendix B.1). It is correct that zero is a trivial solution that satisfies the smoothness requirement yet contains no useful information. Therefore the optimal solution is only non-trivial when the objective is a combination of $\ell_2$ approximation and Laplacian regularization. Under such scenarios, the node features might be noisy themselves, and the Laplacian regularizer serves as a way to use neighborhood features as a (dynamic) denoiser.
>
>
> **Q2: What are nice properties of GHiPPO solutions?**
> The GHiPPO solutions are desirable in the following aspects:
>
> - It allows a linear dynmaical system (LDS) representation under specific choice of approximation configurations, which is a nice property of HiPPO framework that is inherited by GHiPPO. Without a reasonable ODE representation, the approximation framework would have been only technically of interest but impractical.
> - ODE representations allow a constant memory update rule via utilizing a suitably defined discretization procedure. This ensures that the inference time memory and computational complexity is well-controlled.
> - GHiPPO also inherits the versatility of HiPPO in that the LegS configuration is not the only one that allows ODE parameterization. Indeed we may use alternative approximation schemes such as fourier approximation and the resulting framework is still efficiently computable and practical.
>
> **Q3: The semantics of discretization**
>
> This is a very good question, and we think this is an important algorithmic challenge raised by the GHiPPO abstraction. In our framework, we view the underlying temporal graph as a collection of node feature processes together with a dynamically-changing topological relation among them. Under this framework, we regard graph snapshots as discrete observations of this underlying graph process. The objective of discretization is thus to develop a node state update rule that respects the graph dynamics, which is the central topic we discussed in section 3.2 of our paper. In particular, the existence of dynamic topological relations brings significant challenges to the discretization scheme (note that without topological relations, we can directly apply the ZOH discretization rule in a node-wise fashion) due to unobserved mutations of the underlying temporal graph which is conceptually depicted in equation (7) and figure 2 in our paper. Therefore, to study what should be a good discretization in this case, we first establish a dicretization rule in hindsight (the oracle discretization in theorem 2) and propose approximations to the oracle discretization thereafter.
>
> [3] DyGFormer: Towards Better Dynamic Graph Learning: New Architecture and Unified Library, NeurIPS 2023
>
> [4] SimpleDyG: On the Feasibility of Simple Transformer for Dynamic Graph Modeling, WWW 2024
>
> ---
> Thank you again for taking the time to review our paper. We hope our responses could clarify your concerns, and hope you will consider increasing your score. If we have left any notable points of concern unaddressed, please do share and we will attend to these points.

---

> > ### Comment · Reviewer_PUCW · 2024-08-09
> >
> > I would like first to thank the authors for their diligent response. However, I still found my main concerns not fully addressed after reading the response.
> >
> > 1. I don't think "they focus on modelling continuous graph" is a sufficient reason for not comparing them with the proposed method. Continuous VS discrete are often modelling choices most of the time, and memory-based TGNN have been applied to the dataset used in the paper [1,2]. I also think the modelling choice of this paper is also assuming the underlying process is continuous?
> >
> > The most appealing reason why I think a detailed comparison is necessary between memory-based TGNN and the proposed method is their memory mechanism. Memory-based TGNN captures dynamic/temporal information of the graph by incrementally processing information and distilling it into a so-called memory vector, which in a sense can be viewed as the state of the graph/node. Therefore, I think the mechanisms between memory-based TGNN and the method proposed are very similar. I am very keen to understand the fundamental difference and why we need this potentially new family of models for dynamic graphs.
> >
> > 2. I am very confused with the choice of experimental setting for the link prediction task. Based on my understanding, ROLAND actually focused on the efficiency of TGNN and studied a learning setting that is different from the typical ones (e.g., [1,2]). They studied and proposed a framework that is related to continual/incremental learning.
> >
> >
> > [1] Huang, Shenyang, et al. "Temporal graph benchmark for machine learning on temporal graphs." Advances in Neural Information Processing Systems 36 (2024).
> >
> > [2] Poursafaei, Farimah, et al. "Towards better evaluation for dynamic link prediction." Advances in Neural Information Processing Systems 35 (2022): 3

---

> > > ### Author Response · Authors · 2024-08-14
> > > **Gentle Reminder**
> > >
> > > Dear Reviewer PUCW,
> > >
> > > We sincerely appreciate your insightful review and feedback comments.  As the author-reviewer discussion deadline (Aug 13) is approaching, we would like to check if you have any other remaining concerns about our paper.
> > >
> > > **We have faithfully responded to ALL your comments. If our responses have adequately addressed your concerns, we kindly hope that you can consider increasing the score.** We understand that you have a demanding schedule, and we appreciate the time and effort you dedicate to reviewing our paper.
> > >
> > > Kind regards,
> > >
> > > Authors

---

> > > > ### Comment · Reviewer_PUCW · 2024-08-14
> > > >
> > > > Thank you for the diligent response. I have raise my score accordingly.

---

> > > > > ### Author Response · Authors · 2024-08-14
> > > > > **Thank you for raising the score**
> > > > >
> > > > > We are very happy to hear that you have raised the score. Your recognition of our efforts in the paper is greatly appreciated! We thank you again for the invaluable feedback and insights.

---

> ### Author Response · Authors · 2024-08-12
> **Response to Reviewer PUCW (Part 1/2)**
>
> Thanks for acknowledging our efforts in the rebuttal. We greatly appreciate your prompt response during the rebuttal period. We will now address your additional concerns and questions below.
>
> **Response to Q1:** Thank you for the question regarding TGNNs. In our initial rebuttal, due to limited characters we only tried to answer your question in a somewhat conventional way by pointing out modeling on discrete-time dynamic graphs (DTDG) and continuous-time dynamic graphs (CTDG) are usually considered as different tasks[3]. We agree with you that there are more profound connections between the modeling of DTDG, CTDG and the GHiPPO abstraction we proposed. Our perspectives are three-fold:
>
> + (i) We will show that DTDG and CTDG are two different lossy observation schemes of the underlying graph process in the GHiPPO abstraction.
> + (ii) We discuss why the current SSM framework based on discretizations are not yet readily suitable for handling CTDGs.
> + (iii) We return to CTDG models as you have mentioned, and present some empirical results suggesting that they might indeed be not an ideal solution to DTDG problems.
>
> ### (i) DTDG, CTDG and GHiPPO
>
> Recall that in our formulation of the underlying graph process, the node features evolve continuously and the topological relations among nodes allow finite (countable) mutations. Let's rewrite the process in equation (2) of the paper:
> $G(0) \overset{\mathcal{E}_1}{\longrightarrow} G(t_1) \overset{\mathcal{E}_2}{\longrightarrow} G(t_2) \longrightarrow \cdots \longrightarrow  G(t\_{M-1}) \overset{\mathcal{E}_M}{\longrightarrow}  G(t_M) = G(T).$
>
> Now we take a closer look at DTDG and CTDG representations of the above process:
>
> - **DTDG**: In DTDG representations, we do not directly observe the events, but we observe the entire graph at certain time spots resulting in a serious of snapshots. In this spirit, **DTDG has complete latitudinal information, but is lossy regarding longitudinal information**.
> - **CTDG**: In CTDG representations, we have complete observations of events, but upon each event information, we do not observe the features of the rest of the nodes (that do not participate in those specific events). Therefore, **CTDG has complete longitudinal information, but is lossy regarding latitudinal information**.
>
> Next we proceed to why we choose to only study the former one (DTDG).
>
> ### (ii) Handing CTDGs using SSM discretizations is challenging
>
> In section 3.2 of our paper (especially theorem 2), we established the discretization scheme upon an ideal, discrete observation. (We observe the graph snapshot at each mutation events) We believe that this one reasonably hints tha gap between possible empirical approximations in either DTDG or CTDG scenarios. In DTDGs, we believe approximations using available snapshots are possible since from hindsight, the ideal representation is a convex combination of the snapshot representations at the mutation times. The approximation bias mostly comes from fewer snapshots, and we use mixing strategies to mitigate the biases.
>
> However, in CTDG scenarios, we miss the majority of information in each snapshot. Up till now we have not yet figured out any practical and sound solutions to this issue. Besides, consturcting snapshots from CTDGs is itself a very impractical method. It is possible that the GHiPPO abstraction might not be a very good fit for a principled study of CTDGs (might be too stringent, in our opinion). Hence, we regard the modeling of CTDG to be beyond the scope of GraphSSM.
>
> ### (iii) CTDG models and their performance on DTDG graphs
>
> As you have pointed out, many CTDG models are based on a **Message Passing combined with Recurrent State Update (MPRSU)** scheme, which is closely related to GraphSSM (and indeed, related to many other DTDG methods like ROLAND as well). Recall that, SSMs are also a special case of linear RNNs from the computational viewpoint. However, **SSMs utilize a finer-grained definition of memory (via deriving them from an online approximation problem) that yields more stable and performant RNN variants as compared to GRU or LSTM.** This is indeed what we tried to establish in our paper---We equip the node-wise memory in a principled way that is formalized as the optimal approximation coefficients of the Laplacian-regularized online approximation problem. We have shown that this formulation leads to better DTDG modeling frameworks, and in our ablation study we also show the efficacy of using SSM schemes rather than ordinary RNNs or even softmax transformers. So far as we have noticed, most TGNN methods have no mechanistic interpretations of their memory mechanisms. Besides, as we have discussed earlier, it is unclear if SSM schemes are readily applicable to CTDG frameworks.
>
> Finally, we have gained some additional empirical insights from additional experiments. We conducted evaluations of CTDG methods on the DBLP-3, Brain, Reddit, and DBLP-10 datasets.

---

> ### Author Response · Authors · 2024-08-12
> **Response to Reviewer PUCW (Part 2/2)**
>
> The comparison CTDG baselines include memory-based methods TGN and TIGER (as you mentioned in the initial review), as well as attention-based methods TGAT and DyGFormer. Results for the node classification tasks are presented below:
>
> ||**DBLP-3**||**Brain**||**Reddit**||**DBLP-10**||
> |---------|:-----------:|:-----------:|:-----------:|:-----------:|:-----------:|:-----------:|:-----------:|:-----------:|
> ||Micro-F1|Macro-F1|Micro-F1|Macro-F1|Micro-F1|Macro-F1|Micro-F1|Macro-F1|
> |TGN|80.43|80.08|89.50|89.59|40.66|40.50|70.01|68.83|
> |TIGER|81.53|81.74|90.49|90.33|41.17|40.98|72.48|71.14|
> |TGAT|82.59|82.14|90.98|90.25|42.79|42.40|73.15|72.02|
> |DyGFormer|84.77|84.51|92.01|91.32|45.24|44.51|72.40|71.86|
> |GraphSSM|**86.29±1.0**|**85.78±0.9**|**93.80±0.3**|**94.47±0.6**|**49.21±0.5**|**49.05±0.7**|**76.80±0.3**|**76.00±0.4**|
>
> Due to time and space limitation, we are currently presenting the experimental results of the node classification task on the four datasets. Further experiments are being conducted and can be presented in a few days if it is necessary. As observed, CTDG methods demonstrate relatively poor performance on the four datasets, which is inline with our above claims. Indeed, many CTDG methods require fine-grained time information to capture the evolution patterns between edges, which is impractical for DTDGs with coarsened snapshot information.
>
> **Response to Q2:** Firstly, we acknowledge that the learning setting in ROLAND differs from that of other works [1, 2]. The main differences arise from diverse contexts of dynamic graphs, as detailed below:
>
> + ROLAND focus on the **discrete-time dynamic graphs (DTDG)**, which is also the main focus of our work. In this setting, our goal is to predict edges in snapshot 𝑡 + 1 based on the snapshots accumulated up to time 𝑡. This is the general setting in discrete-time temporal graphs and is widely used in literature including several representative works, e.g., EvolveGCN[4].
> + For TGB[1] and DGB[2], both benchmarks focus on the **continuous-time dynamic graphs (CTDG)**, whose goal is to predict the occurenc of a link between two given nodes at a specific time point.
> + While we agree that "Continuous vs. discrete are often modeling choices most of the time", there are differences in terms of the associated time information in the graphs for modeling. In the context of CTDG, link prediction is **fine-grained**, with each link being associated with an exact timestamp. This allows for predicting evolving edge streams at specific time points. However, in DTDG, link prediction is **coarse-grained**. Each edge in a snapshot has relative time information, and we can only predict whether a link will occur in the next time span, such as in the next week or month. In this regard, we were unable to perform link prediction under the setting of [1,2].
>
> Secondly, while ROLAND studied and proposed a framework that is related to continual/incremental learning, the experimental setup for the link prediction task exactly follows the classic DTDG settings in previous works[4,5,6,7]. This is detailed in the ROLAND paper ("Task" section in Section 4.1).
>
> Finally, to avoid any potential confusion, we detail our experimental setting for the link prediction task:
>
> + **Train-val-test splits:** For each dataset, we use $G_1, G_2, · · · , G_{T −1}$ as the training data, $G_{T}$ as test graph. Also, we use $G_{T-1}$ as the validation graph during training for hyperparameter tuning. Each model leverages the T-1 visible graphs to prediction the occurence of edges in $G_T$.
> + **Positive and negative edges:** For the link prediction problem, the positive test set consists of the edges that appear in $G_T$ and do not present in $G\_{T-1}$, while the negative test set consists of the edges that do not appear in $G\_{T-1}$ and $G_T$.
> + **Evaluation metrics:** AUC and AP, two standard metrics in link prediction tasks.
> ---
>
> If any concern still remains that might prohibit a positive recommendation of this work, we would appreciate if you could let us know.
>
>
> ### Reference
>
> [3] Kazemi, Seyed Mehran, et al. "Representation learning for dynamic graphs: A survey." Journal of Machine Learning Research 21.70 (2020): 1-73.
>
> [4] Pareja, Aldo, et al. "Evolvegcn: Evolving graph convolutional networks for dynamic graphs." Proceedings of the AAAI conference on artificial intelligence. Vol. 34. No. 04. 2020.
>
> [5] You, Jiaxuan, Tianyu Du, and Jure Leskovec. "ROLAND: graph learning framework for dynamic graphs." *Proceedings of the 28th ACM SIGKDD conference on knowledge discovery and data mining*. 2022.
>
> [6] Zhang, Kaike, et al. "Dyted: Disentangled representation learning for discrete-time dynamic graph." *Proceedings of the 29th ACM SIGKDD Conference on Knowledge Discovery and Data Mining*. 2023.
>
> [7] Sankar, Aravind, et al. "Dysat: Deep neural representation learning on dynamic graphs via self-attention networks." Proceedings of the 13th international conference on web search and data mining. 2020.

---

### Official Review · Reviewer_o7Uk · 2024-07-15

**Soundness:** 3
**Presentation:** 3
**Contribution:** 3
**Rating:** 6
**Confidence:** 2

**Summary:**

This paper investigates SSM theory to temporal graphs by integrating structural information into the online approximation with laplacian regularization term.

**Strengths:**

1. The proposed method has the theoretical support to show the effectiveness of the proposed method.
2. The experimental results show that the proposed method outperform most of the baseline methds.
3. This paper is well-motivated.

**Weaknesses:**

1. Since one advantage of the SSM based methods is its small parameter size, then what's the size of the model compared with other baseline methods? It's better to list the number of parameters to show the efficiency of the proposed method.

2. The transformer based methods have better performance as shown in Table 1, do you include any transformer-based methods for the experimental comparison?

**Questions:**

1. Since one advantage of the SSM based methods is its small parameter size, then what's the size of the model compared with other baseline methods?

2. The transformer based methods have better performance as shown in Table 1, do you include any transformer-based methods for the experimental comparison?

3. What is the time complexity of the GraphSSM?

**Limitations:**

The authors provide the limitation in appendix E.

---

> ### Author Rebuttal · Authors · 2024-08-06
>
> We appreciate your positive reviews. Your concerns are addressed as follows:
>
> **W1 & Q1: Parameter size of GraphSSM and baseline models**
>
> Thank you for your insightful suggestion. Per your suggestion, we have now included the comparison of parameter sizes between GraphSSM models and baselines on a large dataset arXiv.
>
> |       | STAR | tNodeEmbed | EvolveGCN | SpikeNet | ROLAND | GraphSSM-S4 | GraphSSM-S5 | GraphSSM-S6 |
> | ----- | ---- | ---------- | --------- | -------- | ------ | ----------- | ----------- | ----------- |
> | arxiv | OOM  | OOM        | 2.5M      | 780K     | 1.2M   | 294K        | 93k         | 229K        |
>
> As observed from the table, GraphSSM, particularly GraphSSM-S5, shows superior parameter efficiency compared to other methods. This is attributed to the advantages offered by SSM models.
>
> **W2 & Q2: Comparison of transformer-based methods.**
>
> In this work, we focus on **discrete-time temporal graphs**, which typically have long sequence lengths and are not suitable scenarios for transformers. Current works on transformer-based temporal graph learning mainly focus on **continuous-time temporal graphs**, leaving a significant gap for discrete-time temporal graphs. In this regard, we did not and cannot include transformer-based baselines in our experiments. And it is unfair to directly apply continuous-time methods to our specific settings for comparison. Therefore, we conduct an ablation study on GraphSSM by substituting the SSM architecture with Transformer and present the results below.
>
> |                      |  **DBLP-3**   |               |   **Brain**   |               |  **Reddit**   |               |  **DBLP-10**  |               |
> | -------------------- | :-----------: | :-----------: | :-----------: | :-----------: | :-----------: | :-----------: | :-----------: | :-----------: |
> |                      | **Micro-F1**  | **Macro-F1**  | **Micro-F1**  | **Macro-F1**  | **Micro-F1**  | **Macro-F1**  | **Micro-F1**  | **Macro-F1**  |
> | GraphSSM-Transformer |   85.02±1.1   |   84.98±0.8   |   93.47±1.5   |   93.11±1.1   |   43.48±0.7   |   43.11±0.5   |   75.65±0.8   |   74.32±0.6   |
> | GraphSSM-S4          |   85.26±0.9   |   85.00±1.3   |   93.52±1.0   |   93.54±0.9   | **49.21±0.5** | **49.05±0.7** | **76.80±0.3** | **76.00±0.4** |
> | GraphSSM-S5          | **86.29±1.0** | **85.78±0.9** |   93.00±0.4   |   93.01±0.4   |   44.75±0.4   |   44.79±0.4   |   75.19±0.6   |   73.95±0.4   |
> | GraphSSM-S6          |   86.10±0.5   |   85.70±0.6   | **93.80±0.3** | **94.47±0.6** |   43.11±0.9   |   42.85±1.1   |   74.09±0.3   |   73.16±0.2   |
>
> As observed, GraphSSM-Transformer does not exhibit significantly superior performance in learning from discrete graph sequences compared to SSM-based architectures, despite introducing more memory and computation overheads.
>
> **Q3: Time complexity of the GraphSSM**
>
> GraphSSM is a discrete-time sequence model that performs a message passing step for each graph snapshot, followed by an SSM model for sequence learning. Assuming we have $L$ GNN layers and the sequence length is $T$, the full-batch training and inference time complexity for each graph snapshot with $L$ layers can be bounded by $\mathcal{O}(L|\mathcal{E}|d)$, which represents the total cost of sparse-dense matrix multiplication or message passing. We assume a hidden dimension of $d$ across all layers for simplicity.
>
> |                         |                                  |
> | ----------------------- | -------------------------------- |
> | Graph message passing   | $\mathcal{O} (TL\| \mathcal{E}\| d)$ |
> | Graph sequence learning | $\mathcal{O}(T)$                 |
>
> The main bottleneck of GraphSSM is the graph message passing step, which can be further optimized using techniques such as graph sampling[1] or graph condensation[2].
>
> [1] GraphSAINT: Graph Sampling Based Inductive Learning Method. ICLR 2020.
>
> [2] Graph Condensation for Graph Neural Networks. ICLR 2022.
>
> ---
>
> We would be grateful if you tell us whether the response answers your questipns of GraphSSM, if not, what we are lacking, so we can provide better clarification. Thank you for your time.

---

> > ### Comment · Reviewer_o7Uk · 2024-08-11
> > **Reply to authors' rebuttal**
> >
> > I would like to thank the authors for the detailed response. I don't have further question.

---

> ### Author Response · Authors · 2024-08-11
>
> Thank you for your response. We greatly appreciate your thoughtful feedback and the time you dedicated to reviewing our paper!

---

### Decision · Program_Chairs · 2024-09-25

**Decision:**

Accept (poster)

**Comment:**

This paper presents a well-motivated and technically sound approach to applying state space models (SSMs) to temporal graph neural networks (TGNNs), supported by strong theoretical foundations and promising experimental results. However, it has some limitations, such as the lack of comparison with recent memory-based TGNNs, which are state-of-the-art, and the omission of key tasks like link prediction in the empirical evaluation. Additionally, the model's applicability is limited to discrete-time graphs, and potential issues with heterophilic graphs warrant further exploration. Despite these drawbacks, the paper offers a valuable and interesting research direction.